# Human iPSCs-based modeling unveils SETBP1 as a driver of chromatin rewiring in GATA2 deficiency

Joan Pera [1,2,3], Damia Romero-Moya [1,2], Eric Torralba-Sales [1,2], Rebecca Andersson [2,4], Violeta García-Hernández [5,6,22], Maria Magallon-Mosella[1,2], Maximiliano Distefano[1,2], Clara Berenguer Balaguer[7], Julio Castaño [8], Francesca De Giorgio[1,2], Zhichao Qiu[1,2,9,10], Arnau Iglesias [5,6,11], Paulina Spurk[12,13], Sara Montserrat-Vazquez[2,4], Lorenzo Pasquali[7], Zhuobin Liang [9], Albert Català[14,15], M. Carolina Florian [2,4,16,17], Marcin W. Wlodarski [18], Anna Bigas[5,6,11], Oskar Marin-Bejar [1,2,19] ✉ & Alessandra Giorgetti [2,16,20,21] ✉

Patients with GATA2 deficiency are predisposed to developing myelodysplastic neoplasms (MDS), which can progress to acute myeloid leukemia. This progression is often associated with cytogenetic and somatic alterations. Mutations in SETBP1 and ASXL1 genes are recurrently observed in GATA2 patients, although their roles remain poorly understood. Here we develop a hiPSC-based system to investigate the impact of SETBP1 and ASXL1 mutations in GATA2 deficiency. Using precise genome editing, we recreate stepwise mutational trajectories observed in GATA2-related MDS. We demonstrate that GATA2 mutation has limited impact on hematopoietic progenitors, while the co-occurrence of SETBP1 or ASXL1 mutations impairs myeloid differentiation. The combination of all three mutations severely depletes myeloid progenitors, recapitulating GATA2-related MDS and highlighting their synergistic interplay. Notably, SETBP1 mutation plays a dominant role in establishing a stable chromatin accessibility landscape, even when co-occurring with ASXL1. Our study establishes an iPSC-based model of GATA2 deficiency, offering new insights into myeloid disease progression and a platform for testing future therapeutic strategies.

GATA2 is a master regulator of hematopoietic development, highly expressed in hematopoietic stem cells (HSCs) and early hematopoietic progenitors[1,2]. Its essential role in hematopoiesis was highlighted by knock-out studies in mice, which die *in utero* from severe anemia[3,4]. Notably, mouse Gata2-null endothelial cells failed to produce HSCs because of impaired endothelial to hematopoietic transition (EHT)[2,3,5]. We and others have also demonstrated a primary role of GATA2 in promoting EHT in a human setting[6,7]. In humans, germline heterozygous GATA2 mutations result in pleiotropic manifestations with hematologic cytopenia leading to myelodysplastic neoplasms (MDS); immunodeficiency in multiple cell lineages involving B, NK, monocytic, CD4+, and dendritic cells; and can also lead to deafness and lymphedema[8–10]. In accordance with existing literature and our empirical observations, it has been established that at least 80% of individuals carrying GATA2 mutations develop MDS at an estimated age of 17–20 years[11–15], which may subsequently progress to acute myeloid leukemia (AML)[16].

GATA2 deficiency is a well-established predisposing factor for pediatric MDS, accounting for ~7% of all pediatric MDS cases and up to

15% of cases with excess blasts[13,17]. The clinical presentation of GATA2 deficiency is heterogeneous, with considerable variability in both expressivity and disease severity. Despite this phenotypic diversity, the condition exhibits high lifetime penetrance for immunodeficiency and MDS[13,15]. Interestingly, some individuals, even within the same family, may remain asymptomatic or maintain normal hematological parameters throughout life, suggesting either incomplete penetrance or delayed disease onset[11,13,17]. Life expectancy is limited mainly because of the occurrence of immunodeficiency, bone marrow failure (BMF), MDS, or AML[8,13,16,18]. The only curative option is allogenic hematopoietic stem cell transplantation (HSCT), usually performed when patients progress to advanced MDS or severe immunodeficiency.

The mutational landscape of GATA2 deficiency includes a steadily increasing number of variants with ~800 documented cases, revealing around 180 different familial and de novo GATA2 germline mutations[15,19]. Four major types of GATA2 mutations have been identified: truncating mutations, missense mutations (mostly located in the zinc finger domain 2), noncoding mutations, particularly those disrupting the auto-enhancer site within intron 4, and synonymous GATA2 mutations causing monoallelic RNA degradation[14,20].

To date, only a limited number of GATA2 mutations have undergone comprehensive functional analysis. Overall, missense mutations exhibit potential impairment in DNA binding capability, whereas others indicate haploinsufficiency mechanisms[21]. MDS and leukemia arise as secondary events in GATA2 carriers, with cytogenetic and/or somatic events playing a significant role in malignant transformation and delineating disease prognosis. Monosomy 7 is the most common cytogenetic aberration in children with GATA2 deficiency[13], while the prevalence of trisomy 8 increases with age[12,17]. A consensus on somatic mutations in genes such as SETBP1, ASXL1, and STAG2 has recently been established in GATA2-related MDS patients[11,12,22]. However, the complete elucidation of the pathophysiology underlying GATA2 deficiency remains elusive.

Although the development of genetically engineered animal models for selected GATA2 mutations has provided significant insights on GATA2 deficiency, the understanding of the molecular mechanisms driving malignant transformation is challenging due to the lack of a suitable human model. Recently, human induced pluripotent stem cells (hiPSCs) have emerged as a promising avenue for investigating adult MDS and leukemogenesis[23–25].

In this study, we combined hiPSCs with CRISPR/Cas9 technology to investigate the role of SETBP1 (D868N) and ASXL1 (G646Wfs*12) mutations in the progression of myeloid transformation in GATA2 deficiency. We generated a series of isogenic human iPSC lines harboring single, double, and triple combinations of patient-relevant mutations. Our results show that GATA2 haploinsufficiency leads to mild hematopoietic defects. In contrast, both SETBP1 and ASXL1 independently impair myeloid differentiation in the context of GATA2 deficiency. Strikingly, the combination of all three mutations results in a severe depletion of myeloid progenitors, closely recapitulating the hematopoietic phenotype observed in GATA2-related MDS patients[13]. Transcriptomic and epigenetic analysis provide mechanistic insights into disease progression and unveil a key epigenetic role of SETBP1 mutation. Our study establishes a platform to investigate not only additional somatic mutations reported in GATA2-deficient patients but also potential epigenetic dysregulation contributing to disease pathogenesis.

## Results

### SETBP1 and ASXL1 mutations impaired differentiation of GATA2-mutant cells

We previously generated GATA2 mutant hiPSC lines by introducing the monoallelic R396Q missense mutation (loss-of-function; LoF) into the endogenous GATA2 locus of a healthy hiPSC line[26], using CRISPR/Cas9 technology.

Here, to investigate the contribution of secondary genetic events in GATA2 deficiency, we generated a series of isogenic human iPSC lines harboring single, double, and triple mutations. Specifically, we introduced both SETBP1 (D868N gain-of-function; GoF) and ASXL1 (G646Wfs*12 LoF) mutations in heterozygosity by CRISPR/Cas9 technology (Fig. 1A). The SETBP1 and ASXL1 mutations were confirmed by Sanger sequencing (Supplementary Fig. 1A). We selected these two mutations based on previous studies[27,28] and our recent analysis of the largest cohort of GATA2-MDS patients to date[17]. In this study, we analyzed 218 individuals with GATA2 deficiency and identified SETBP1 and ASXL1 as the most frequent somatic mutations in GATA2-related MDS, predominantly in association with monosomy 7, confirming our previous data[13]. In line with prior studies[23], our efforts to model monosomy 7 in hiPSCs were met with significant technical challenges, underscoring the complexity of recapitulating this chromosomal alteration in vitro. Therefore, we focused our study on SETBP1 and ASXL1 mutations. Next, two independent clones from each condition (P=parental (Isogenic control), A=ASXL1, S=SETBP1, G=GATA2, GA=GATA2-ASXL1, GS=GATA2-SETBP1 double mutant, and GSA=GATA2-SETBP1-ASXL1 triple mutant) were selected and characterized. All mutant hiPSC lines showed normal karyotype and expressed pluripotent stem cell markers (Supplementary Fig. 1B, C). To evaluate the impact of GATA2 mutation, both alone or in combination with driver mutations, on hematopoietic development, we performed in vitro myeloid differentiation using a well-established embryoid bodies (EB)-based protocol[7]. This protocol promotes mesoderm induction, the specification of mesodermal cells into hemogenic endothelial progenitors (HEPs), and the subsequent commitment of HEPs to hematopoietic progenitor cells (HPCs) (Supplementary Fig. 1D). Flow cytometry analysis on day 15 of EB differentiation revealed mild hematopoietic alterations in the G condition with an increase of monocytes (CD33+CD14+) (Fig. 1B–G), resembling the early manifestation of the disease previously reported in multiple GATA2-deficient patients who developed MDS and carried monosomy 7[13,19,29]. However, this monocytosis appears transient, likely due to the lack of monosomy 7 in our model system.

SETBP1 mutation alone promotes an expansion of early HSPCs. However, this increase is blunted by the presence of GATA2 mutation, resulting in an impairment of myeloid lineage differentiation. While ASXL1 mutation alone does not impair hematopoietic differentiation, its effect becomes pronounced in the context of GATA2 deficiency, revealing a context-specific role (Fig. 1B–G). The combination of all three mutations led to a more severe phenotype, characterized by the loss of HSPCs as well as mature CD33+CD11b+ myeloid lineages (Fig. 1F), with a marked reduction of monocytes (CD14+) (Fig. 1G), but no difference in neutrophils (CD15+) (Fig. 1H). Consistent with the decreased hematopoietic output, GA and GSA hiPSC lines produced limited colony formation in methylcellulose culture (Fig. 1I). The compromised myeloid differentiation observed in GSA mirrors the differentiation defects observed in individuals with GATA2 deficiency[15]. To assess whether the impaired hematopoietic development observed in our mutant hiPSCs stemmed from a reduced proliferation/survival rate of myeloid progenitors, we analyzed the distribution of cell cycle and apoptosis in CD33+ cells on day 15 of EB development. Flow cytometry analysis showed an increase in apoptosis and a significant decrease in cycling cells in both GA and GSA conditions (Fig. 1J and Supplementary Fig. 1E), indicating that myeloid HPCs have less proliferation capability and are activated by a stress response. In summary, our findings reveal that, within the context of GATA2 deficiency, mutations in SETBP1 and ASXL1 impair myeloid differentiation in vitro. This disruption is most pronounced in the triple-mutant lines, which exhibit a profound depletion of myeloid progenitors, underscoring a synergistic interaction among these somatic mutations that drive hematopoietic dysfunction.

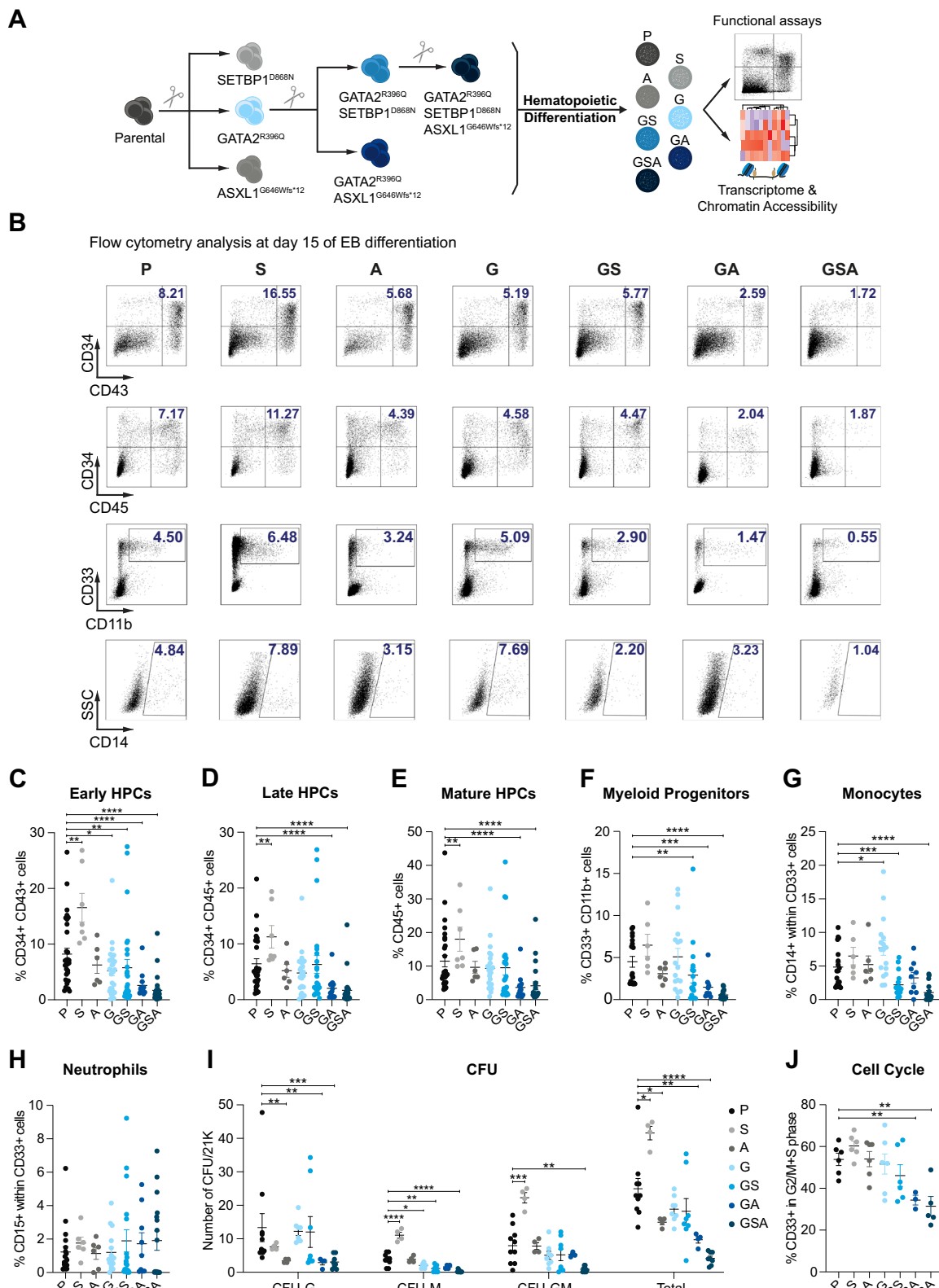

SETBP1 drives stable chromatin remodeling in GATA2-deficient cells

To gain insights into the consequence of GATA2, SETBP1, and ASXL1 mutations on the chromatin landscape, we performed ATAC-seq on CD34+CD43+CD33+CD45+ hiPSC-derived cells, sorted from isogenic control (P) and mutant hiPSC lines (G, S, A, GA, GS, and GSA) (Supplementary Fig. 2A). Principal component analysis (PCA) based on

chromatin accessibility data showed that S, GS and GSA conditions clustered together as well as A and GA, distinguishing them from P and G conditions (Fig. 2A). The cumulative count analysis of differentially accessible peaks showed a typical peak annotation, with more than half of the peaks falling into enhancer regions (intronic and intergenic regions) (Supplementary Fig. 2B). Moreover, to comprehensively investigate chromatin dynamics across successive

**Fig. 1 | SETBP1 and ASXL1 mutations impair in vitro hematopoietic differentiation of GATA2-mutant cells. A** Schematic overview of mutant-hiPSCs generation by CRISPR/Cas9 and characterization. **B** Representative flow-cytometry analysis illustrating early (CD34+CD43+) and late (CD34+CD45+) hematopoietic progenitors, myeloid progenitors (CD33+CD11b+), and monocytes (CD14+) on day 15 of differentiation in P, S, A, G, GS, GA, and GSA conditions. **C** Quantitative summary of CD34+CD43+ cells at day 15 of differentiation. Data represent the mean ± SEM of independent biological replicates (P=32, S=7, A=6, G=26, GS=26, GA=13, and GSA=26). **D** Quantitative summary of CD34+CD45+ cells at day 15 of differentiation. Data represent the mean ± SEM of independent biological replicates (P=29, S=7, A=6, G=23, GS=23, GA=13, and GSA=23). **E** Quantitative summary of total CD45+ cells at day 15 of differentiation. Data represent the mean ± SEM of independent biological replicates (P=29, S=7, A=6, G=23, GS=23, GA=13, and GSA=23). **F** Quantitative summary of total CD33+CD11b+ cells at day 15 of differentiation. Data represent the mean ± SEM of independent biological replicates (P=21, S=6, A=6, G=17, GS=17, GA=9, and GSA=17). **G** Quantitative summary of CD14+ within CD33+ cells at day 15 of differentiation. Data represent the mean ± SEM of independent biological replicates (P=18, S=6, A=6, G=17, GS=17, GA=9, and

GSA=17). **H** Quantitative summary of CD15+ within CD33+ cells at day 15 of differentiation. Data represent the mean ± SEM of 6-18 independent biological replicates (P=18, S=6, A=6, G=16, GS=17, GA=9, and GSA=17). **I** Colony-forming unit (CFU) potential of day 15 EB-derived hematopoietic progenitors. Colonies were scored on morphology as CFU-granulocyte(CFU-G), CFU-macrophage(CFU-M), and CFU-granulocyte-macrophage(CFU-GM). Data represent the mean ± SEM of the total number of colonies per 21,000 cells seeded, compiled from independent biological experiments (P=10, S=4, A=4, G=8, GS=8, GA=4, and GSA=8). **J** Cell-cycle analysis of CD33+ cells at day 15 of differentiation using EdU-incorporation and DAPI-staining. Data represent the mean ± SEM of independent biological replicates (P=6, S=6, A=6, G=6, GS=6, GA=3, and GSA=5). P=Parental, S=SETBP1 mutant, A=ASXL1 mutant, G=GATA2 mutant, GS=GATA2-SETBP1 mutant, GA=GATA2-ASXL1 mutant, GSA=GATA2-SETBP1-ASXL1 mutant. Statistical analysis: Data with a normal distribution were analyzed using the two-sided Student's *t*-test, while non-normally distributed data were analyzed using the two-sided Mann–Whitney test. Statistical significance was indicated as follows: *p < 0.05* (*), *p < 0.001* (**), and *p < 0.0001* (****). Source data are provided as a Source Data file. Exact *p*-values can be found on source data.

stages, we conducted a comparative analysis of chromatin alterations in each condition (G, S, A, GA, GS, and GSA) relative to the chromatin landscape of P. The representation of the chromatin accessibility changes of the different experimental conditions in a heatmap showed the same trend of the PCA plot, suggesting that distinct mutation profiles are responsible for shaping different patterns of chromatin accessibility (Supplementary Fig. 2C). Minimal alterations of chromatin accessibility were evident in G juxtaposed with P. Interestingly, most of the changes in chromatin accessibility observed in the A condition were preserved in GA but were largely lost in GSA. In contrast, the chromatin accessibility changes were more similar between S and GS, and even more concordant between GS and GSA. These findings suggest that the changes induced by SETBP1 mutation have a more sustained impact, it dominates in shaping stable chromatin accessibility landscapes in the context of GATA2 deficiency (Supplementary Fig. 2C), as also visualized by the tornado plot (Fig. 2B). Next, we performed an in-depth analysis of ATAC-seq data to identify regulated genes based on chromatin accessibility and the magnitude of changes. To uncover the chromatin changes driving the observed phenotypes in our mutant hiPSCs, we analyzed differentially accessible peaks (DAPs) across A, S, GA, GS, and GSA conditions versus P. In order to specifically assess the contribution of S and A mutations within the context of GATA2 deficiency, we excluded peaks that were unique to A and S, and not maintained in the combined GA, GS, or GSA conditions. To dissect key chromatin accessibility changes, we prioritized peaks with an absolute log2FC > 0.5 and adjusted *p*-value < 0.05[30] (Fig. 2C and Supplementary Fig. 2D, and Supplementary Table 1). We detected 4073 peaks with increased accessibility in the GS condition compared to P (Fig. 2C). Notably, the majority of these (*n* = 3073) remained accessible in the subsequent GSA stage. In contrast, only 1311 of the 9037 peaks gained in the GA condition were maintained in GSA, consistent with our previous observations. A similar trend was observed for regions with reduced chromatin accessibility; most closed DAPs in GA were not preserved in GSA, whereas a substantial fraction of the closed regions identified in GS persisted into the GSA stage (Fig. 2C). Gene set enrichment analysis (GSEA) of genes associated with open DAPs in GA, GS and GSA revealed distinct functional enrichments such as epithelial-mesenchymal transition (EMT) and cell morphogenesis (Fig. 2D). Interestingly, the SETBP1 mutation has a stronger impact on chromatin remodeling, progressively priming hematopoietic stem cells for leukemic transformation by increasing accessibility at key regulatory loci. These changes first appeared in GS and were maintained in GSA (Fig. 2D). Conversely, several genes related to myeloid differentiation and hematopoietic development were consistently downregulated in GA, GS and GSA (Fig. 2D). These

data suggest that, in the context of GATA2 deficiency, both ASXL1 and SETBP1 mutations contribute to an impairment of myeloid differentiation.

Notably, some of the accessible chromatin regions gained in GS and GSA, but not in GA, have been linked to genes involved in hematological disorders, such as *MEF2C*, whose upregulation is associated with poor outcome in pediatric AML[31], the leukemogenic drivers *MEIS1*[32], *HOXB3*, known to promote cell growth in pre- and established leukemia[33]; *ETV6, HOXB9, LMO2* and *MEIS2*, transcription factors, commonly associated with AML[34,35] (Figs. 2E, 3A; Supplementary Figs. 3A, B and 4A). Genes related to HSC leukemic pathway founded in GS and GSA included also *BAALC* and *HLF*[36,37]. The activated genes common in GA and GSA included *CALD1, LAMA3* and *NRP1* involved in cell adhesion, cellular migration and morphogenesis[38–40]. Furthermore, *MECOM*, a SETBP1 direct target whose upregulation has been previously associated with SETBP1 GoF mutation[41] and BMF[42]; *CDK6*, that acts as a transcriptional regulator to suppress EGR1[43] and it is usually overexpressed in AML cells; *ETS1*, involved in granulocyte differentiation and found overexpressed in patients with chronic myeloid leukemia[44], were found open in all conditions (Figs. 2E, 3A; Supplementary Figs. 3B and 4A). Moreover, the advanced GSA stage showed specific open chromatin accessibility regions, mapping to *HOXA3*, *HOXB5* and *HOXB7* genes involved in HSC and myeloid progenitor maintenance, blocking the differentiation when overexpressed[45–47] (Figs. 2E, 3A S3B). Conversely, among the 618 less accessible genes identified in the overlapping analysis of GS, GA and GSA (Figs. 2E, and 3B), we observed key regulators of hematopoietic development and myeloid differentiation, including *GFI1B*[48], *ITGA2B*[49], and *EGR1*[43], *NOTCH1* and *CREBBP*[50], as well as *CUX1*, a tumor suppressor frequently lost in the MDS[51] (Fig. 3B and Supplementary Fig. 4B). Other genes crucial for the function of hematopoietic progenitors, such as *GATA1*[52] and *WNT3/WNT4*, were downregulated earlier in GS and later GSA stage[48,53]. The expression of genes involved in DNA damage response and cell cycle were downregulated in both GA and GSA, likely contributing to impaired proliferation of myeloid progenitors, as observed in vitro.

Furthermore, we compared the chromatin accessibility profiles of P, G, GS, GA and GSA hiPSC-derived HPCs with those of primary human hematopoietic cell types along the hematopoietic hierarchy[54]. The chromatin landscapes of P, G, and GA conditions resembled those of normal hematopoiesis. On the other hand, GS/GSA chromatin profiles are most different from the monocytic lineage but not any of the progenitor populations (Pearson correlation: 0.8–1) (Supplementary Fig. 4C). These observations suggest a chromatin priming of GS and GSA conditions toward more multipotent progenitors, which may lead to a myeloid differentiation blockage, in line with what we have

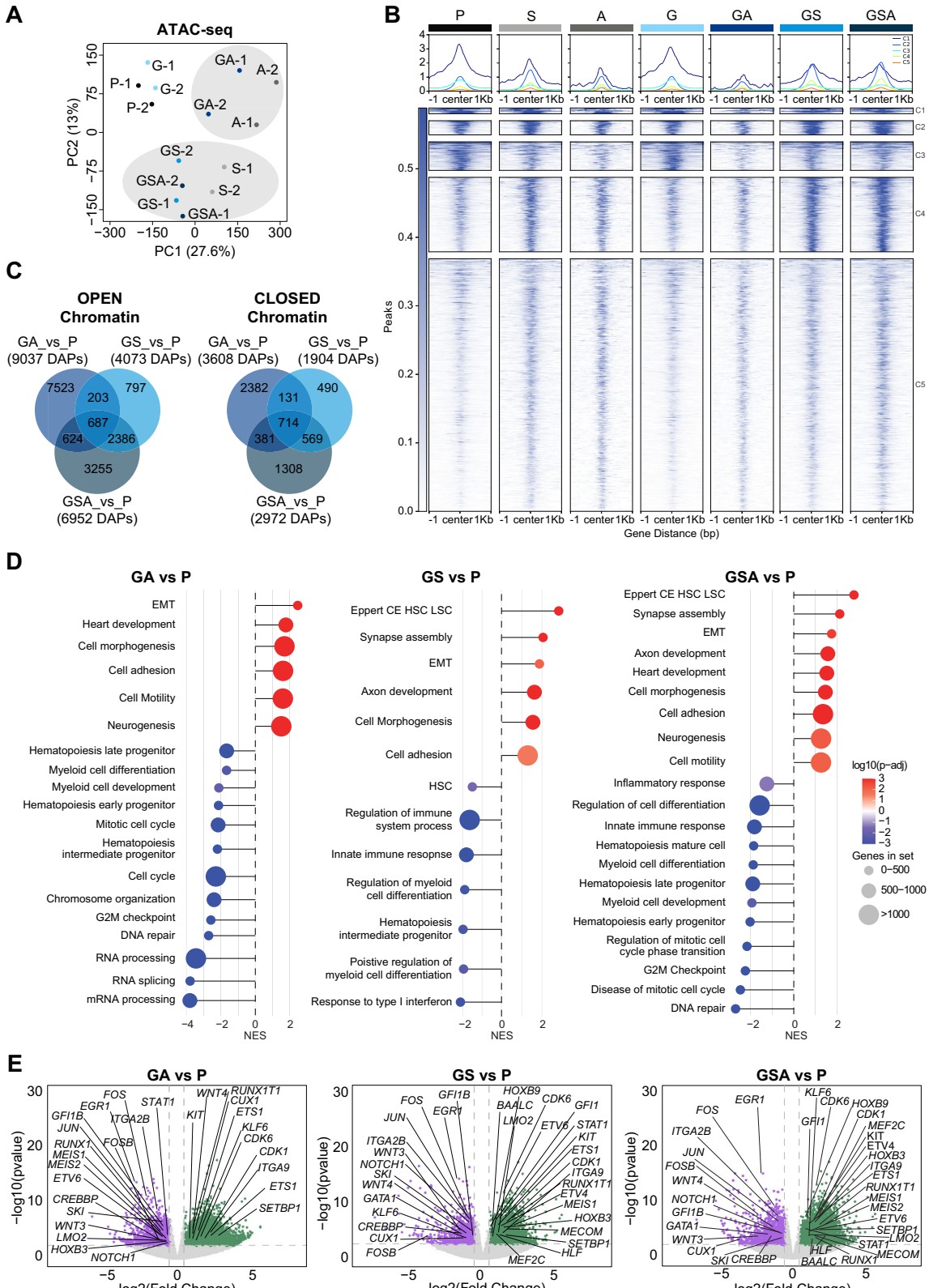

observed in vitro blood differentiation experiments of Fig. 1. To validate the role of SETBP1 in driving chromatin remodeling in GATA2 deficiency, we compared previously published SETBP1 ChIP-seq data[55] with our bulk ATAC-seq profiles from GS and GSA conditions. This analysis revealed altered accessibility at SETBP1 target genes such as *MECOM*, *MEF2C*, and *HOXB3* (Supplementary Fig. 4D), supporting the notion that SETBP1 contributes substantially to stable epigenetic

reprogramming, even in the presence of additional mutations like ASXL1.

Finally, to confirm that the chromatin accessibility changes were real and not due to mixed cell population, we employed Simultaneous High-throughput ATAC and RNA Expression with sequencing (SHARE-seq)[56], that allows the single-cell profiling of low number cells at unprecedented resolution. We sorted CD34+CD33+CD43+CD45+ cells

**Fig. 2 | SETBP1 mutation leads to stable chromatin remodeling in GATA2 deficient cells. A** Principal-component analysis (PCA) of bulk ATAC-seq data depicting multidimensional distribution of chromatin accessibility of different samples. **B** Tornado plot representation of bulk ATAC-seq differential accessible peaks (DAPs) across P, S, A, G, GA, GS, and GSA conditions. Each line corresponds to a distinct DAP, with the signal plotted within a representation of the peaks signal profile within a ±1 kb window centered on the peak summit. Signal coverage is illustrated using a color gradient ranging from white (no coverage) to blue (maximum coverage), reflecting chromatin accessibility at each condition. Tornados are divided in five clusters (C1-C5) based on peaks similarities. **C** Venn diagrams showing overlap of open and closed DAPs in the indicated comparisons. **D** Gene set enrichment analysis (GSEA) of the chromatin accessibility profiles comparing GA vs P, GS vs P, and GSA vs P conditions. The lollipop plots highlight significant down-regulation of inflammatory response and hematopoietic differentiation pathways in GS and GSA, alongside upregulation of in leukemic stem cell associated gene signatures in GSA. All gene sets shown have a False Discovery Rate (FDR) < 0.05,

indicating statistically significant enrichment. GSEA was performed with the clusterProfiler R package. Significance of enrichment scores was evaluated by permutation testing (two-sided), with multiple testing correction by the Benjamini–Hochberg false discovery rate (FDR). **E** Volcano plots showing differential chromatin accessibility in GS vs P, GA vs P, and GSA vs P comparisons. Each point represents an individual ATAC-seq peak, with the log2FC > 0.5 in accessibility (x-axis) plotted against the −log10(p-value) (y-axis). Peaks associated with, defined by p-value < 0.05 and absolute log2FC > 0.5, are colored. Genes associated with significant DAPs are indicated. Differential chromatin accessibility was assessed using edgeR and limma. Count data were modeled with a negative binomial distribution, and moderated t-tests were performed. Reported p-values are two-sided and adjusted for multiple comparisons using the Benjamini–Hochberg false discovery rate (FDR). P=Parental, S=SETBP1 mutant, A=ASXL1 mutant, G=GATA2 mutant, GS=GATA2-SETBP1 mutant, GA=GATA2-ASXL1 mutant, GSA=GATA2-SETBP1-ASXL1 mutant.

from five experimental conditions (P, G, GA, GS, and GSA). After de-multiplexing of the sequencing data, we applied a conservative threshold and 141,900 cells with >1000 alignments were retained for downstream analysis. Following quality filtering based on standard metrics, we obtained single-cell chromatin accessibility profiles for 16,270 cells. Dimensionality reduction revealed a largely uniform cluster, lacking discrete clusters (Supplementary Fig. 5A). To validate this result, we quantified the Shannon entropy of each cell, which demonstrated that all conditions exhibited a comparable chromatin accessibility profile (Supplementary Fig. 5B). We further assessed cluster robustness using the Silhouette width, a metric that evaluates how well an individual cell fits to its assigned cluster. The consistently low values observed across all conditions (Supplementary Fig. 5C), indicating that our dataset is comprised of a largely homogenous cell population with minimal epigenetic heterogeneity. Then, differential accessibility analysis comparing G, GA, GS and GSA conditions versus P identified a total of 986 DAPs and 474 associated genes (Supplementary Table 2). Given the relatively low number of DAPs and the limited resolution of the scATAC-seq data, likely reflecting technical constraints of the method, we integrated the single-cell ATAC-seq dataset with the original bulk ATAC-seq data using a pseudo-bulk approach (nf-core/atacseq pipeline) to validate and strengthen these findings (Supplementary Table 3). As shown in Supplementary Fig. 5D–F, this comparison revealed strong concordance between pseudobulk and bulk ATAC-seq datasets, validating our previous findings. This indicates that the observed chromatin accessibility differences are not attributable to cellular heterogeneity within sorted populations but instead represent condition-specific regulatory changes occurring in a predominantly homogeneous cell population.

## Accessible regions enriched for transcription factor networks in myeloid malignancies

In order to gain deeper insights into the gene regulation networks underlying the observed changes in chromatin accessibility, we performed transcription factors (TF) motif enrichment analysis within peaks exhibiting differential accessibility (either open or closed) employing the Hypergeometric Optimization of Motif EnRichment (HOMER) method. ATAC peaks containing AP-1 motifs (JUN, FOS and BATF) gained accessibility in A and GA condition but were subsequently lost in GSA (Fig. 3C and Supplementary Fig. 6A). On the other hand, ETS motifs were already enriched in S and were maintained in GS and GSA, while HOX and RUNX motifs emerged in GS and were also maintained in GSA, suggesting potential involvement of these TFs in the hematopoietic impairment (Fig. 3C and Supplementary Fig. 6A).

Notably, all these TFs are linked to normal hematopoietic development, and their dysregulation is associated with several myeloid neoplasms[35,57]. Conversely, GATA motifs[58], were highly enriched in closed chromatin regions across S, GA, GS, and GSA (Fig. 3C), but not in

A conditions (Supplementary Fig. 6B). Importantly, integration of ATAC-seq data with a previously published GATA2 ChIP-seq dataset[59] revealed that, among the 714 closed chromatin regions shared by GA, GS and GSA, 229 regions (32%) correspond to genes directly targeted by GATA2 (Fig. 3D and Supplementary Table 4). Notably, these genes include *NOTCH1, ITGA2B*, and *EGR1*. On the other hand, only 18 of 687 (2.6%) open chromatin regions common to GA, GS, and GSA conditions were direct GATA2 targets (Fig. 3D). These data have been further validated employing Find Binding Motif Occurrences (FIMO) analysis (Supplementary Table 5). Overall, these results suggest that the acquisition of secondary somatic mutations potentially impacts the GATA2 gene regulatory network, leading to the loss of chromatin accessibility at GATA2 target genes in the GSA conditions.

## Transcriptomic signature predicts myeloid impairment in GSA-hiPSCs

Next, to identify transcriptomic changes at different stages, we performed RNA-seq analysis on purified CD34+CD43+CD33+CD45+ cells at day 15 of blood differentiation from P, A, S, G, GA, GS, and GSA hiPSC lines. PCA revealed distinct clustering patterns, with S, GS, and GSA conditions grouping together, while P, G, A, and GA samples formed separate clusters (Fig. 4A). Notably, GATA2 mutation alone produced minimal transcriptomic changes relative to P, consistent with the ATAC-seq observations (Supplementary Fig. 7A). Therefore, our analysis specifically examined differentially expressed genes (DEGs) across GA, GS, and GSA conditions, excluding changes unique to either S and A conditions alone. After the application of cutoffs (log2FC > 0.5, adj. p-value < 0.05), we identified a total of 1190 DEGs; upregulated (GA vs P n = 179, GS vs P n = 203 and GSA vs P n = 208) and downregulated (GA vs P n = 104, GS vs P n = 287 and GSA vs P n = 209) (Fig. 4B, Supplementary Fig. 7B, and Supplementary Table 6). Interestingly, most transcriptomic changes observed in GA were not maintained in GSA, consistent with the corresponding chromatin accessibility patterns from our ATAC-seq analysis (Fig. 4B and Supplementary Fig. 7A–C). Importantly, correlation analysis reveals significant concordance between observed transcriptional alterations and corresponding chromatin accessibility changes, with RNA-seq and ATAC-seq profiles showing 73% correlation in GA, 85% in GS, and 82% in GSA conditions (Supplementary Fig. 7C).

Consistent with our ATAC-seq findings, the most prominent transcriptomic change was the sustained downregulation of hematopoietic and myeloid differentiation gene signatures. This suppression first appeared in GS condition and remained stable through the GSA condition. These results reinforce the notion of impaired blood differentiation. In parallel, we observed an upregulation of leukemic stem cell gene signatures in both GS and GSA, suggesting features of disease progression and/or exhaustion of myeloid progenitors[60] (Fig. 4C, D). To further extend our transcriptome analysis, we compared our RNA-

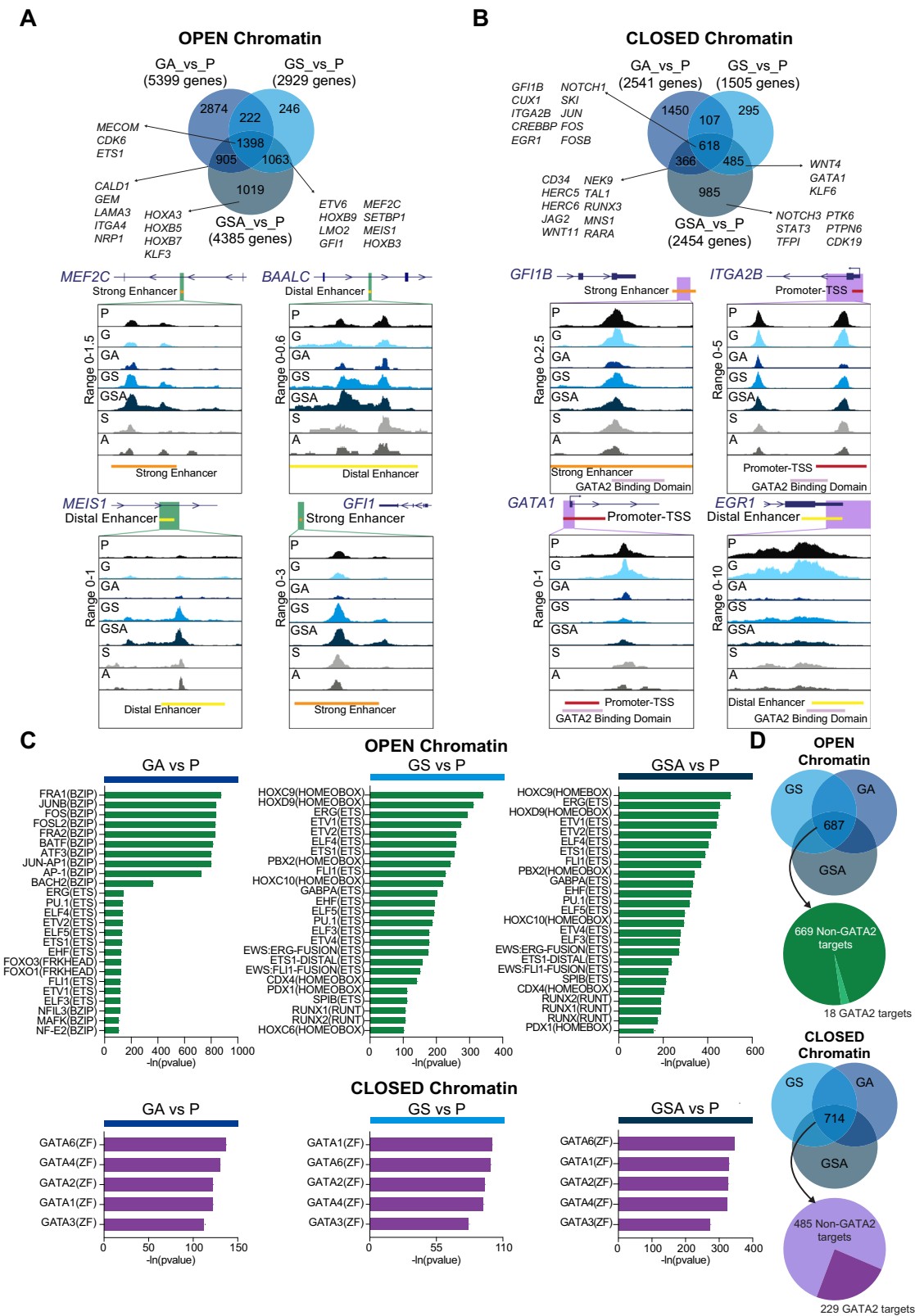

seq datasets with publicly available transcriptomic data from iPSC lines generated from MDS patients with germline GATA2 mutation (MDS3.1 and MDS3.4) and their isogenic control (H1.1)[23]. Notably, GS and even more prominently GSA mutant hiPSCs exhibited a transcriptomic profile closely resembling that of the reprogrammed cells from high-risk GATA2-related MDS patients, consistent with an MDS-like phenotype (Fig. 4E). These results reinforce a dominant impact of SETBP1

mutation on the transcriptomic landscape of mutant GATA2 cells. Overall, our comprehensive analysis suggests that SETBP1 initiates epigenetic and transcriptional reprogramming within early myeloid progenitors in the context of GATA2 deficiency. The concomitant acquisition of ASXL1 mutations further reinforces the differentiation impairment, ultimately contributing to the establishment of an MDS phenotype. Importantly, our hiPSCs-based model faithfully

**Fig. 3 | Chromatin accessibility landscape of iPSC-derived HSPCs. A** Venn Diagram and ATAC-seq tracks illustrating open differentially accessible peaks (DAPs), defined by *p*adj<0.05 and abs(log2FC > 0.5). Tracks for *MEF2C, BAALC, GFI1* and *MEIS1* loci. Display of the genomic interval exhibiting the highest logFC from the comparison of condition GSA versus condition P. ATAC-seq signal is color-coded by condition: P (black), G (light-blue), GS (blue), GA (navy-blue), and GSA (dark-blue), S (light-grey), and A (dark-grey). Annotated regulatory elements are indicated: strong-enhancer (orange), distal-enhancer (yellow) and promoter-TSS (red), Gen-ehancer track of UCSC Genome Browser(hg38). Green-box marks genomic interval displayed for each gene locus. Differential chromatin accessibility was assessed using edgeR and limma. Count data were modeled with negative binomial distribution, and moderated *t*-tests. Reported two-sided *p*-values are adjusted for comparisons using Benjamini–Hochberg false discovery rate (FDR). **B** Venn Diagram illustrating closed DAPs, defined by *p*adj<0.05 and abs(log2FC > 0.5). Tracks for *GFI1B, GATA1 and ITGA2B, ERG1* loci. ATAC-seq signal is color-coded by condition: P (black), G (light-blue), GS (blue), GA (navy-blue), and GSA (dark-blue), S (light-grey), and A (dark-grey). Annotated regulatory elements are indicated: strong-enhancer (orange), distal-enhancer (yellow) and promoter-TSS (red), Gen-ehancer track of UCSC Genome Browser(hg38). Purple-box marks genomic interval displayed for each gene locus. Differential chromatin accessibility was assessed using edgeR and limma. Count data were modeled with negative binomial distribution, and moderated *t*-tests. Reported two-sided *p*-values are adjusted for comparisons using Benjamini–Hochberg FDR. **C** (Top) HOMER motif analysis of open DAPs. Top 25 enriched transcription factor (TF) binding motifs are shown for DAPs across GA (left), GS (middle), and GSA (right) conditions versus P. Each motif is annotated by corresponding TF name, family and enrichment in *p*-value, as determined by HOMER. (Bottom) HOMER motif analysis of closed DAPs. Top 5 enriched TF binding motifs are shown for DAPs across GA (left), GS (middle), and GSA (right) conditions versus P. Each motif is annotated by corresponding TF name, family and enrichment in *p*-value, as determined by HOMER. Motif enrichment analysis was performed with HOMER (knownMotifs). Enrichment *p*-values were calculated using one-sided cumulative hypergeometric test, with comparisons corrected by Benjamini–Hochberg FDR. Source data are provided as a Source Data file. **D** GATA2-targets identified by ChIP-seq data reported by Fujiwara et al.[59] crossed with DAPs in the GA, GS, and GSA intersection. (Top) Overlapping 18 out of 687 genes with GATA2-bound targets. (Bottom) Overlapping 229 out of 714 genes with GATA2-bound targets. P=Parental, G=GATA2 mutant, GS=GATA2-SETBP1 mutant, GA=GATA2-ASXL1 mutant, GSA=GATA2-SETBP1-ASXL1 mutant.

## Discussion

Individuals carrying GATA2 mutations are prone to develop immunodeficiency and BMF, which can progress to MDS and AML. Previous studies using various animal models, including mouse and zebrafish[61–68], have investigated the molecular mechanism underlying GATA2 deficiency. While these studies have significantly advanced our understanding of some aspects of GATA2 deficiency, the existing models fail to fully replicate the multilineage dysplasia observed in GATA2 carriers. Furthermore, none of them have examined the impact of additional somatic mutations commonly found in GATA2-deficient patients. In this study, we developed a humanized iPSC-based model to investigate the role of secondary oncogenic lesions in the context of GATA2 deficiency. Our model revealed that the acquisition of secondary mutations in *SETBP1* (GoF) and *ASXL1* (LoF) led to an impairment of hematopoietic differentiation, inducing a marked decrease of myeloid progenitors. Our in vitro model revealed that heterozygous GATA2 mutations alone are insufficient to induce an MDS phenotype in a human context, consistent with previous studies[23,69]. These data reinforce the concept that additional genetic events are required to drive disease progression in individuals with germline GATA2 mutations. This is in line with clinical observations showing that newborns with germline GATA2 mutations typically do not exhibit hematological manifestations until the first decade of life, when secondary genetic alterations emerge. Notably, more aggressive phenotypes in childhood are often associated with cytogenetic alterations such as monosomy 7[13,17]. A limitation of our study was the inability to reproduce monosomy 7 in the mutant hiPSCs, consistent with previous studies[23]. This absence may explain the lack of leukemic transformation in our in vitro model.

Despite this limitation, our work provides the first human model to study the functional impact of secondary oncogenic mutations in the context of GATA2 deficiency. Phenotypic analysis revealed that both ASXL1 and SETBP1 mutations independently impair myeloid differentiation in the contest of GATA2 deficiency, with a synergistic effect observed when they co-occur at more advanced stages. Through molecular analysis of hiPSCs-derived myeloid progenitors across different conditions, we identified distinct epigenetic and transcriptomic alterations. Importantly, in the context of GATA2 deficiency, the acquisition of the SETBP1 mutation appeared to exert a more dominant impact than ASXL1, as the chromatin accessibility changes observed in the GS condition were largely preserved in GSA, whereas those induced by ASXL1 in GA were mostly lost in the presence of SETBP1.

The majority of SETBP1 mutations, such as p.D868N, hinder its ubiquitination and subsequent degradation, resulting in increased SETBP1 protein levels in the cell[70]. While the contribution of increased SETBP1 expression to myeloid transformation has been corroborated by several studies[71,72], its role in the context of GATA2 deficiency remains unexplored. Our ATAC-seq analysis provided two key insights concerning the impact of SETBP1 mutation in a GATA2-deficient background. First, SETBP1 activation led to increased chromatin accessibility at genes implicated in MDS pathogenesis and enrichment of a leukemic HSC signature, including transcription factors such as *BAALC* and *HLF*[36,37]. Second, this was accompanied by the down-regulation of genes essential for stem cell self-renewal and myeloid commitment.

ASXL1 encodes a chromatin regulator[73], and mutations in this gene can induce chromatin remodeling that reflects transient epigenetic noise rather than stable reprogramming. Notably, the few stable alterations driven by ASXL1 include key regulators of myeloid differentiation and cell cycle control, many of which are also targeted by SETBP1, suggesting that these changes may be sufficient to account for the phenotypes observed in our in vitro model.

Interestingly, motif enrichment analysis revealed that genomic regions with loss of chromatin accessibility were enriched in GATA motifs, indicating that SETBP1 and ASXL1 mutations significantly reduce accessibility to potential GATA2 targets. Importantly, this observation was validated by integrating our ATAC-seq data with publicly available GATA2 ChIP-seq dataset[59].

Emerging data support the concept that SETBP1 mutations represent the earliest somatic oncogenic events in GATA2 patients[17,74]. ASXL1 mutations are also frequently observed in GATA2-related myeloid malignancies, and are thought to occur as subsequent events following SETBP1 mutations[17,22,74,75]. Of note, a study of 386 patients with MDS demonstrated that SETBP1 mutation enhances the expression of mutant ASXL1 and plays a pivotal role in the leukemic transformation from MDS to AML[76]. Whether SETBP1 directly activates ASXL1 or cooperates with it in disease progression, potentially leading to impaired myeloid differentiation, remains an open question and warrants further investigations.

Furthermore, the inactivation of inflammatory pathways correlates with the impairment of myeloid differentiation and the strong reduction of monocytes. It has been previously described that monocytopenia leads to an altered response to cytokines, such as IFN-

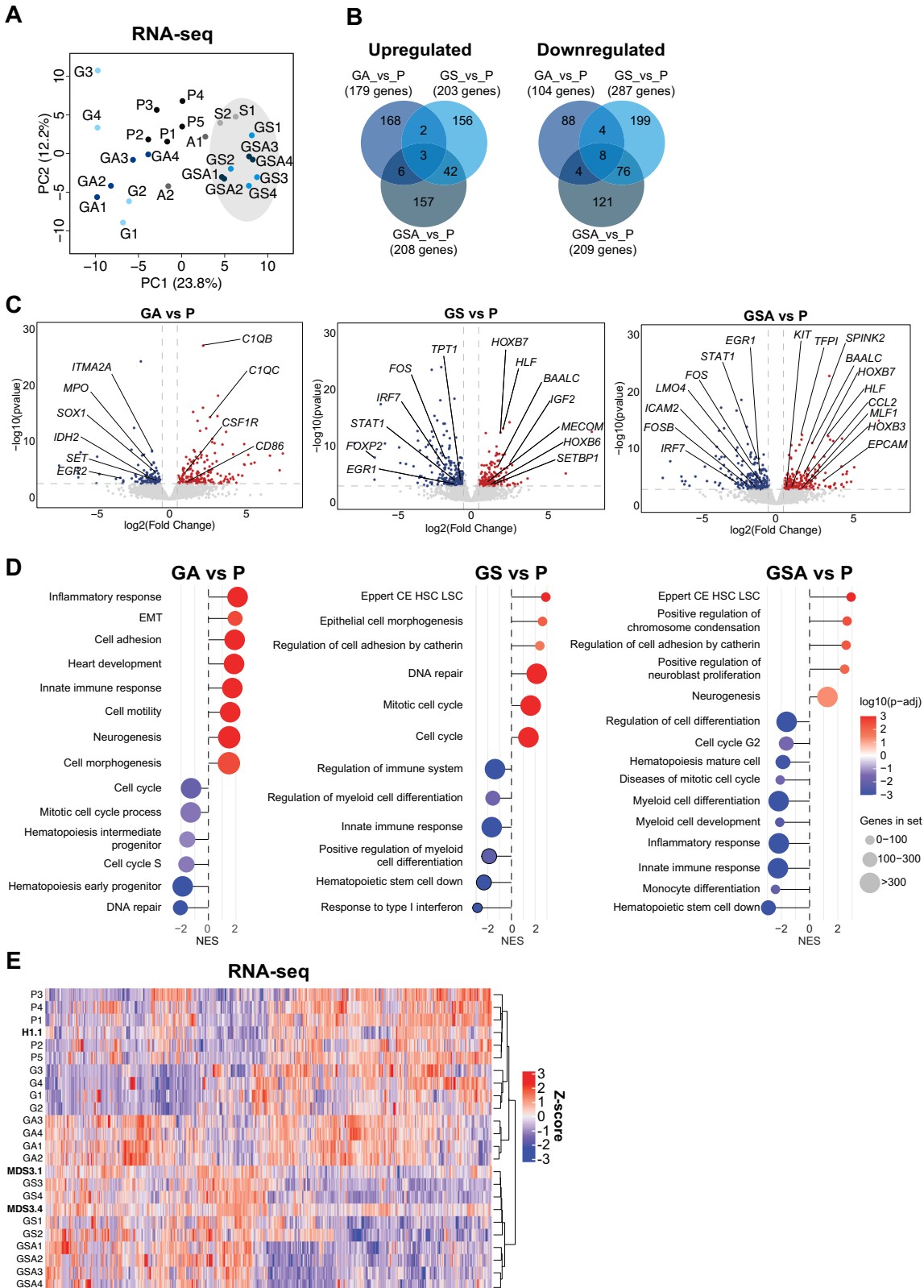

γ, IL-12, IL-6, and TNF-α, predisposing individuals to recurrent infections. Therefore, our results are in line with the heightened susceptibility to bacterial infections documented in patients with GATA2 deficiency[19]. In support of our results indicating SETBP1 as the driver of MDS phenotype in our model, we demonstrate similarities in the gene expression profiles between myeloid progenitors derived from our GS and GSA mutant hiPSCs and those derived from high-risk GATA2-related MDS hiPSCs[23]. In conclusion, our hiPSC-based system represents the first humanized model of GATA2 deficiency that enables the study of endogenously acquired oncogenic mutations, uncovering key chromatin alterations that drive MDS-associated pathology.

This work sheds light on the mechanisms driving MDS in GATA2 deficiency and point to SETBP1-related pathway as potential targets for future therapies. Nonetheless, further investigations are warranted to

**Fig. 4 | Transcriptomic dynamics across specific stages of GATA2 deficiency.**
**A** Principal component analysis (PCA) of RNA-seq data across differentiation conditions. PCA was performed on transcriptomic profiles from seven experimental conditions: P, S, A, G, GA, GS, and GSA. Each point represents an individual biological replicate, with clustering reflecting global gene expression similarities and differences among conditions. **B** Venn diagram illustrating the overlap of differentially expressed genes (DEGs) across indicated comparisons. Upregulated and downregulated genes were identified based on RNA-seq analysis, and their intersections are shown to highlight shared and condition-specific transcriptional responses. **C** Volcano plots showing differential gene expression in GA vs P, GS vs P, and GSA vs P comparisons. Each point represents a gene, with log2FC > 0.5 in expression plotted on the x-axis and −log(p-value) on the y-axis. Significantly DEGs are colored, illustrating transcriptomic changes across differentiation stages relative P. Differential gene expression analysis was performed using DESeq2. Significance was assessed with the Wald-test (two-sided) on negative binomial model. p-values were adjusted using the Benjamini−Hochberg false discovery rate (FDR). **D** Gene set enrichment analysis (GSEA) of the transcriptomic profiles, comparing GA vs P, GS vs P, and GSA vs P conditions. The lollipop plots highlight significant

downregulation of inflammatory response and hematopoietic differentiation pathways in GS and GSA, alongside upregulation of in leukemic stem cell associated gene signatures in GSA. All gene sets shown have a False Discovery Rate (FDR) < 0.05, indicating statistically significant enrichment. GSEA was performed with the clusterProfiler R-package. Significance of enrichment scores was evaluated by permutation testing (two-sided), with correction by the Benjamini−Hochberg FDR. **E** Heatmap displaying hierarchical clustering of samples based on gene expression profiles. Shown are expression levels of all genes that were differentially expressed (p-adj<0.05) in at least one of the pairwise comparisons. The analysis includes integration of our dataset with RNA-seq data from reprogrammed iPSCs derived from myelodysplastic neoplasm (MDS) patients MDS3.1 and MDS3.4, and healthy donor H1.1, as reported in Kotini et al.[23]. Clustering reveals transcriptomic relationships across experimental and reference samples. Differential gene expression analysis was performed using DESeq2. Significance was assessed with the Wald test (two-sided) on a negative binomial model. P-values were adjusted for multiple comparisons using the Benjamini−Hochberg false discovery rate (FDR). P=Parental, S=SETBP1 mutant, A=ASXL1 mutant, G=GATA2 mutant, GS=GATA2-SETBP1 mutant, GA=GATA2-ASXL1 mutant, GSA=GATA2-SETBP1-ASXL1 mutant.

fully elucidate the clinical significance and implications of these findings.

## Methods

### hiPSCs culture and hematopoietic differentiation

Parental hiPSC-line (CBIO8#4) was generated from CB-derived CD133+ cells using integrative retrovirus. The Spanish National Embryo Ethical Committee approval was obtained to work with hiPSCs (525 428 1). Parental and GATA2-mutant hiPSCs have been banked at the Banco Nacional de Líneas Celulares (BNLC) of ISCIII. hiPSCs were culture on Matrigel (Cultek S.L., #356231) coated plates under feeder-free culture maintenance with mTeSR1 medium (StemCell Technologies, #85850) at 37 °C, 5% $CO_2$ and 20% $O_2$. Medium was changed daily, and cells were passaged weekly by EDTA dissociation (PBS (Fisher Scientific, #10204733) + 0.5 mM EDTA (Becton Dickinson S.A., #347689). For in vitro blood differentiation EBs were cultured from d0 to d2 in mTeSR1 and in the presence of 50 ng/ml BMP4. From d3, EBs were changed to differentiation medium comprising serum-free defined medium (StemPro-34; Invitrogen) supplemented with 0.16 mM monothioglycerol, 15.15 mg/mL holotransferrin, 50 ng/mL BMP-4, 300 ng/mL Fms-related tyrosine kinase 3 ligand (Flt-3L), 300 ng/mL stem cell factor (SCF), 10 ng/mL IL-3 and 10 ng/mL IL-6 (all purchased from R&D Systems). EBs were maintained in culture until d15, with medium replenishment every three days. This established differentiation protocol promotes mesoderm induction (days 2–3), specification of meso-dermal cells to bipotential hemato-endothelial progenitors (HEPs; VE-Cadherin+CD34+CD43-CD45-; days 3–10). These HEPS possess dual differentiation potential towards both endothelial and hematopoietic lineages. The protocol culminates in the commitment of HEPs to hematopoietic progenitor cells (HPCs) (CD34+CD43+CD45+; days 10–15), myeloid progenitors (CD33+CD11b+) monocytes (CD14+ within CD33+), neutrophils (CD15+ within CD33+), recapitulating key milestones of early hematopoiesis.

### CRISPR/Cas9 gene editing

The CRISPR tool (https://bioinfogp.cnb.csic.es/tools/breakingcas/) was used for gRNAs design. A high probability to target the region of interest and low probability to generate off-targets gRNA sequence was selected in each mutation. With exception of the previously generated GATA2 p.R396Q line[26], hiPSCs were prepared for nucleofection by pretreatment with ROCK inhibitor (Y-27632, 10 μM) for 3 h prior to nucleofection. Alt-R® S.p. Cas9 protein (100 pmol; IDT) was complexed with Alt-R® CRISPR-Cas9 sgRNA (120 pmol; IDT) at 25 °C for 10 min, followed by the addition of ssODN (4 μM). A total of $2 \times 10^5$ cells were dissociated using Accutase (Gibco), washed twice with $Ca^{2+}/Mg^{2+}$-free

PBS, and resuspended in 20 μl of P3/S1 buffer. The RNP complex with ssODN was combined with the cell suspension and transferred into a 20 μl cuvette. Nucleofection was carried out using the 4D-Nucleofector System (Lonza) with program CA-137. Cells were then plated in a 12-well dish and cultured in mTeSR1 medium supplemented with Y-27632 (10 μM). After a 72-h recovery period, 1000 cells were seeded at clonal density in a 100-mm dish for single-colony formation. Individual colonies were genotyped by PCR, RFLP, and Sanger sequencing to assess gene editing outcomes[26]. Restriction Fragment Length Polymorphism (RFLP) analysis was performed after PCR with primers Fwd: CAGTCACTTGTGGCGTCTTC (SETBP1) and CAGGACCCTCGCAGACATTA (ASXL1); and Rev: TTCGTGGGCCAGAAAGTTGT (SETBP1) and AGGCGGCAGTAGTTGTGTTC (ASXL1), digestion of the PCR product was done with the restriction enzymes AvaI (SETBP1) and BseRI (ASXL1). The PCR products were analyzed in an agarose gel stained with SYBR-safe (Invitrogen, #S33102). For DNA Sanger sequencing, PCR primers were used.

### Karyotyping

The genomic integrity of the iPSC line was evaluated by G-banded metaphase analysis (300–500 bands) at Sant Joan de Deu, Barcelona. 70% confluent hiPSC colonies were incubated with KaryoMax colcemid (Invitrogen, #15212012), trypsinized, treated with hypotonic solution and fixed in 75% methanol + 25% acetic acid. A minimum of 20 metaphases were examined.

### DNA Sanger sequencing

Genomic DNA was isolated using QIAamp DNA Mini Kit (QIAGEN, #M7806) according to manufacturer's protocol. PCR was performed with the primers, SETBP1-gDNA-PCR and ASXL1-gDNA-PCR (using same primers as mentioned before) with GoTaq Flexi DNA Polymerase (Promega, #M7806) with the following protocol: 95 °C for 5 min, 35 cycles of 95 °C for 30 s, 60 °C for 30 s, 72 °C for 30 s and 72 °C for 7 min. PCR products were purified with DNA Clean & Concentrator-5 (Zymo Research, #D4003) and shipped for sequencing. Genetic alterations were identified using Benchling/Snap Gene.

### Flow cytometry analysis

EBs were treated with collagenase B (Roche, #11088807001) for 1h at 37ºC and dissociated as single cells using PBS + EDTA 0.25mM . Cells were spun at $300 \times g$ for 5 min and washed once with FACS buffer (PBS + 0.5 mM EDTA + 2% HSA (Vitrolife, #10064)). For surface staining, cells were incubated with fluorochrome-conjugated monoclonal antibodies specific to the markers for 15 min at RT in the dark. Following incubation, cells were washed twice with FACS buffer and

resuspended. After flow cytometry sample preparation, cells were resuspended in Annexin V buffer (Immunostep, ANXVF-200T) following manufacturer's protocol. Gallios Flow Cytometer (Beckman Coulter) was used to run samples. The positive population was gated using Kaluza Analysis Software (Beckman Coulter).

For Flow cytometry, following antibodies were used: CD34-PE (Miltenyi Biotec S.L., clone 8G12, #130046702), CD43-APC (Becton Dickinson, clone 1G10, #560198), CD45-APC-H7 (Becton Dickinson, clone 2D1, #560178), CD33-APC (Becton Dickinson, clone WM53, #551378), CD11b-PE-Cy7 (Becton Dickinson, clone ICRF44, #557743), CD14-BV421 (Becton Dickinson, clone MPHIP9, #563743) and CD15-PE (Becton Dickinson, clone HI98, #555402). Dead cell exclusion was achieved using 7-Aminoactinomycin D stain (7AAD) (Invitrogen, #92008). Annexin V-FITC was used to measure apoptosis (Immunostep, #ANXVF-200T).

For Fluorescence activated cell sorting (FACS), following antibodies were used: CD34-FITC (Becton Dickinson, clone 8G12, #348053), CD43-APC (Becton Dickinson, clone 1G10, #560198), CD45-APC-H7 (Becton Dickinson, clone 2D1, #560178), CD33-BV510 (Becton Dickinson, clone WM53, #563257). Dead cell exclusion was achieved using Propidium iodide (Sigma, #P4170-100MG).

To maximize cell recovery, cells were resuspended in 500 μL of FACS buffer. Cells were sorted using MoFlo Astrios EQ, Cell sorter Beckman Coulter instrument. Cells were processed for bulk RNA-seq, bulk ATAC-seq, and scATAC-seq experiments.

### Clonogenic assay
After EBs dissociation, cells were counted and plated in a 12-well plate at a density of 21.000 cells per well in MethoCult (Stemcell Technologies, #H4434). Cells were cultured for 14 days. At day 14, colonies were counted and classified based on morphology as CFU-granulocyte (CFU-G), CFU-macrophage (CFU-M), and CFU-granulocyte-macrophage (CFU-GM).

### Cell cycle assay
At day 15 of blood differentiation EBs were incubated with 10 μM EdU for 7 h, before single-cell dissociation. Single-cell suspension was stained with CD33-APC. Cell proliferation was assessed using the Click-iT Plus EdU cell proliferation kit (Invitrogen, #C10425) following manufacturer's protocol.

### Bulk ATAC-seq of hiPSCs
On day 15 of hematopoietic differentiation, hiPSC-derived myeloid progenitors (CD34+CD43+CD33+CD45+) were sorted, yielding at least 50,000 cells from populations P, S, A, and G, 28,000 from GS, 25,000 from GA, and 8000 from GSA. ATAC-seq library preparations were done following Buenrostro et al.[77] with minor modifications. Briefly, cells were centrifuged for 3 min at 2000 × g at 4 °C and washed once in ice-cold PBS. Then, nuclei were isolated by incubating them in 300 μl cold lysis buffer (10 mM Tris-HCl pH 7.4, 10 mM NaCl, 3 mM MgCl2, 0.1% IGEPAL CA-630) for 20 min and resuspending them after 10 min. The sample was centrifuged for 15 min at 500 × g at 4 °C using low acceleration and brake setting. The resulting pellet was washed with 100 μl of cold lysis buffer and centrifuged again under same conditions. The transposition reaction was carried out in a 25 μl reaction mix containing 1.25 μl of Tn5 transposase (#C01070012, Diagenode), 12.5 μl of 2× tagmentation buffer (#C01019043, Diagenode) and 11.25 μl DEPC-treated water. The reaction mix was incubated at 37 °C for 50 min. To inactivate the reaction, 5 μl of cleanup buffer (900 mM NaCl, 300 mM EDTA), 2 μl of 5% SDS and 2 μl of Proteinase K (#EO0491, ThermoScientific) were added, followed by incubation at 40 °C for 30 min. Tagmented DNA was isolated with 2× SPRI beads cleanup (#A63880, Beckman Coulter) and was eluted in 21 μl 10 mM Tris-HCl pH8.

Two sequential 9-cycle PCRs were performed in order to enrich for small DNA fragments. The PCR mix consisted of 2 μl of 25 μM PCR Primer 1[77], 2 μl of 25 μM Barcoded PCR Primer 2[77], 25 μl of NEBNext High-Fidelity 2× PCR Master Mix (#M0541, NEB) and 21 μl of the eluted sample. The library was amplified in a thermocycler using the following program: 72 °C for 5 min; 98 °C for 30 s; 9 cycles of 98 °C for 10 s, 63 °C for 30 s; and 72 °C for 1 min; and at 4 °C hold. After the first PCR round, fragments smaller than 600 bp were selected using SPRI cleanup beads, 0.6× ratio with right-side selection, and second round of PCR was performed with same conditions. The DNA library was finally purified using 1.8× SPRI cleanup beads, eluting in 21 μl 10 mM Tris-HCl pH8.

The final library was quantified using Qubit dsDNA BR Assay (#Q32850, Invitrogen) and fragment analysis was performed using TapeStation, Agilent Bioanalyzer to check library quality and the nucleosomal pattern. Once library quality was confirmed, samples were sequenced 100–150 bp paired-end to obtain ~70 M reads.

### Bulk ATAC-seq analysis
The ATAC-seq data analysis was performed using the nf-core/atacseq pipeline (10.5281/zenodo.2634132; v2.1.2) from Nextflow with hg38 genome assembly. This pipeline incorporates tools for quality control, read alignment, peak calling, and differential accessibility analysis. Raw reads were pre-processed using Trim-Galore to remove adapter sequences and low-quality bases. Reads were mapped to the hg38 reference genome using the Burrows-Wheeler Alignment tool (BWA, v0.7.17-r1188), and sorted, indexed, and filtered with SAMtools. Normalized coverage was generated using BEDTools (v2.30.0) and bedGraphToBigWig (v445), while peak calling was conducted with MACS2 (v2.2.7.1). Peak annotation was performed using HOMER (v4.11) and consensus peaks were created with BEDTools. Read quantification utilized featureCounts (v2.0.1), and quality control was conducted with MultiQC and FastQC (v0.11.9).

Differential accessibility analysis was performed in R (v4.2.0) using the edgeR (v3.38.1) and limma (v3.52.1) packages. Normalization was carried out using the TMM (Trimmed Mean of M-values) method, and the data were transformed with the voom algorithm to model the mean-variance relationship. A linear model was constructed incorporating both batch and condition as covariates. Empirical Bayes moderation was applied to improve variance estimates across peaks. Differential accessible peaks (DAPs) were identified using moderated t-statistics with Benjamini–Hochberg correction for multiple testing. Peaks were considered significantly differential with an adjusted p-value < 0.05 and an absolute log2FC > 0.5. Dimensionality reduction and visualization were performed using Principal Component Analysis (PCA) and hierarchical clustering heatmaps generated with the ComplexHeatmap (v 2.12.1).

The Visualization of ATAC-seq peak calls and signal tracks was performed using the Integrative Genomics Viewer (IGV), v2.14.0. To contextualize open chromatin regions within the framework of gene regulatory elements, the public "GeneHancer Regulatory Elements" track (v2.18, hg38 assembly This track, which integrates enhancer-promoter interactions from multiple resources, was used to annotate and interpret putative ATAC-seq peaks as potential enhancers or promoters.

Tornado plots were generated using the deepTools (v3.5.6) suite to visualize chromatin accessibility across differential peaks. A ±1 kb window centered on the peak was defined for all DAPs identified in comparisons against the progenitor condition (P). Signal intensity was extracted from normalized BigWig coverage files corresponding to each condition. The resulting matrix was visualized with plotHeatmap, applying k-means clustering (k = 5) to group peaks with similar accessibility profiles. Color mapping was set from white (low signal) to dark blue (high signal).

To assess the similarity of chromatin accessibility profiles between our samples and lineage-specific hematopoietic populations, we incorporated publicly available ATAC-seq data from Corces et al.[54]. This dataset includes bulk ATAC-seq profiles across distinct human hematopoietic lineages, serving as a reference for chromatin landscapes during differentiation. Raw FASTQ files for each population were downloaded from GEO and processed jointly with our bulk ATAC-seq data using the same pipeline and procedures described above. Correlation analysis was performed by calculating Pearson correlation coefficients between the normalized accessibility profiles of our samples and each of the lineage-specific profiles in the Corces dataset, restricted to a shared set of consensus peaks. The resulting correlation matrix was visualized as a heatmap using the R package Complex-Heatmap (v2.12.1), highlighting lineage-specific accessibility patterns and sample similarity.

To assess the GATA2 transcriptional network dysregulation, we integrated publicly available ChIP-seq data from Fujiwara et al. Briefly, from each comparison (GSA vs P, GS vs P, and GA vs P), differentially accessible peaks (DAPs) were classified as regions of gained or lost accessibility relative to progenitors (P). Within each category, we identified regions that were consistently shared across the three comparisons (GSA, GS, and GA), requiring that they exhibited the same accessibility change (gained or lost) in all conditions. These sets of common regions were then intersected with previously mentioned GATA2 ChIP-seq peaks from Fujiwara et al. Overlaps were assessed at the level of genomic coordinates (peak-to-peak overlap), enabling the identification of regions that were both differentially accessible in our dataset and experimentally validated as GATA2 binding sites.

We investigated the presence of GATA2 motifs within differentially accessible chromatin regions using FIMO (Find Individual Motif Occurrences, MEME Suite v5.5.8). We defined two region sets: (i) peaks commonly more accessible in GA, GSA, and GS relative to P, and (ii) peaks commonly less accessible across the same comparisons.

Briefly, all available GATA2 motifs from the JASPAR2024 database were retrieved and combined into a non-redundant motif set after removing duplicates. These motifs were used as input for FIMO motif scanning. Input peak sets were converted to FASTA sequences, and FIMO was run with the default background model and a $p$-value threshold of 1e−4 to retain high-confidence matches. FIMO outputs (motif coordinates, $p$-values, $q$-values, and scores) were intersected with the peak sets to determine which commonly more or less accessible regions contained a GATA2 motif. In addition, we generated a summary table reporting the number of GATA2 motifs identified within each peak containing at least one GATA2 motif (Supplementary Table 5).

## Gene set enrichment analysis (GSEA)
Gene set enrichment analysis (GSEA) was performed using the R package clusterProfiler (v4.10.0), with multiple testing corrections applied via the Benjamini–Hochberg method. For both bulk RNA-seq and bulk ATAC-seq data, genes (or peaks mapped to their nearest gene) were pre-ranked based on the Wald statistic. Enrichment analysis was conducted using curated gene sets obtained from the GSEA Molecular Signatures Database (MSigDB). Gene sets were considered significantly enriched if they had an adjusted $p$-value < 0.05 and included genes with an absolute log2FC > 0.5.

## Motif enrichment analysis
Motif enrichment analysis was performed using the HOMER software suite (v 5.1). Both de novo and known motif analyses were conducted on DAPs identified from each comparison group (G, GS, GSA, GA, S, and A vs P). The findMotifsGenome.pl function implemented in HOMER was used with default parameters to identify statistically enriched transcription factor binding motifs within the DAPs, using the hg38 as assembly reference genome. Background regions were

automatically selected by HOMER to match GC content and length distribution. The resulting enriched motifs were ranked based on their enrichment $p$-value, and both the known motif enrichment and de novo motif discovery results were used for downstream interpretation.

## Single-cell ATAC-seq processing
On day 15 of hematopoietic differentiation, hiPSC-derived myeloid progenitors (CD34+CD43+CD33+CD45+) were sorted, yielding between 32,000 and 40,000 cells from population P; 27,000–31,000 cells from G; 16,000–28,000 cells from GS; 7000–28,000 cells from GA, and 8000 from GSA. The single-cell ATAC-sequencing libraries were produced using the SHARE-seq protocol[56], with the following modifications. All centrifugations were carried out at $500 \times g$ at 4 °C for 5 min in a benchtop centrifuge, if not stated otherwise. Directly after sorting, cells were washed by centrifugation and resuspended at $1 \times 10^6$ cells per mL in PBS with 0.04% BSA, 0.1 U/μL NxGen RNase inhibitor (Biosearch Technologies, cat. no. 30281-2) and 0.025 U/μL SUPERase RNase Inhibitor (Thermo Fisher Invitrogen™ cat. no. AM2696). Cells were fixed with addition of 16% Formaldehyde (Thermo Fischer Pierce™, cat. no. 28906) to a final concentration of 1% (v/v) for 5 min and quenched as described previously.

The transposome was assembled by diluting Tn5 transposase (unloaded) (Diagenode, cat. no. C01070010-20) 1:10 with tagmentase dilution buffer (Diagenode, cat. no. C01070011) and then mixed with an equal volume of annealed transposition adapters (prepared as described in Ma et al. 2020)[60], see Supplementary Table 7 for oligonucleotide sequences) and double the volume of tagmentase dilution buffer, followed by incubation at RT for 30 min. If not used immediately, the transposome was stored for up to 1 h on ice. The tagmentation reactions were carried out with 10,000 cells per reaction, as described[56]. After tagmentation, the samples were centrifuged, the supernatants were discarded and the pelleted nuclei were washed by centrifugation in nuclei isolation buffer (10 mM Tris-HCl pH 7.5/10 mM NaCl/3 mM MgCl2/0.1% NP-40 in Ambion™ Nuclease-free H2O (Thermo Fischer Invitrogen™, cat. no. AM9938)) supplemented with 0.04% BSA, 0.1 U/μL NxGen RNase Inhibitor and 0.05 U/μL SUPERase RNase Inhibitor. The supernatants were removed, and the dry nuclei pellets were frozen by placement in −80 °C and stored until barcode hybridization (for no more than 48 h).

The barcode hybridization was performed as described before[56], see Supplementary Table 7 for all oligonucleotide sequences used for the barcoding steps. Nuclei from each biological replicate were multiplexed together, with each differentiated iPSC cell line indexed by placement in different wells in the first barcoding run to allow for bioinformatic demultiplexing after sequencing. After barcode hybridization, barcode ligation, cell counting and reverse crosslinking, the pre-library mixtures were frozen at −80 °C until library preparation.

The pre-library mixtures were purified using a 1.2× ratio of AMPure Beads (Beckman Coulter, cat. no. A63881) to sample and eluted into 20 μL of Elution Buffer (EB; QIAGEN, cat. no. 19086). The library PCR was performed as described before[60], with the number of cycles established by qPCR. Illumina-compatible i5 indexes were added with the SHARE-seq compatible index primers, see Supplementary Table 7 for their sequences. The finished libraries were cleaned with AMPure Beads at a ratio of 0.8× and eluted into 14 μL elution buffer and stored at −20 °C until use. The libraries were analyzed using the TapeStation 4200 (Agilent) with the High Sensitivity D5000 kit (Agilent, Reagents cat. no. 5067-5593, Screen Tape cat. no. 5067-5592) as per the manufacturer's instructions. Library concentration was determined using the NEBNext® Library Quant Kit for illumina® (New England Biosciences, cat. no. E7630L) qPCR on the QuantStudio 1 RT-PCR (Applied Biosystems). Libraries were pooled at equimolar ratios in a 10 mM Tris-HCl pH 8.3 buffer. Sequencing was performed by Centro Nacional de Análisis Genómico, Barcelona, Spain. Library pool loading concentration was 200pM, and the pool was sequenced on the

NovaSeq 6000 platform (Illumina) with the NovaSeq 6000 S1 1.5v kit (200 cycles) (Illumina, cat. no. 20028318) with the following program: Read 1: 50 cycles, Index 1: 99 cycles, Index 2: 8 cycles, Read 2: 50 cycles.

## Single-cell ATAC-seq analysis

After demultiplexing with a custom script and checking the quality of the reads with FastQC (v0.12.1), raw sequencing reads were trimmed using Cutadapt (v4.9) to remove adapter sequences and low-quality bases. Trimmed reads were then aligned to the reference genome (GRCh38) using Bowtie2 (v2.5.4) with parameters -X 2000 -no-discordant -no-mixed -very-sensitive. Following alignment, BAM files were sorted and indexed using SAMtools (v1.14). Alignments in the mitochondrial chromosome, with quality lower than 30 or overlapping blacklisted regions were removed. Filtered BAM files were turned to BED files using BEDTools (v2.30.0) and PCR duplicates were removed for those reads sharing the same cell barcode, chromosome and start and end position of the alignment. Fragment files were generated from the deduplicated BED file, including Tn5 shifting. Deeptools (v3.5.5) was used to generate BigWig files and to check for Transcription Start Site (TSS) enrichment. R (v4.4.2) was used to plot the fragment size distribution, count and plot the number of fragments per barcode and perform an initial filtering of barcodes with less than 100 fragments.

Fragment files were processed with Signac (v1.14.0) and Seurat (v5.2.1) packages. Fragment objects were created, and peak calling was performed separately for each condition using MACS2 (v2.2.9.1). Peaks were filtered based on empirical thresholds to retain high-confidence regions. A consensus peak list was generated by merging peaks across all conditions and fragment counts were quantified within these regions to build a chromatin accessibility assay and create a Seurat object.

Quality control (QC) metrics were computed for each cell, including TSS enrichment, nucleosome signal scores, and fraction of reads in peaks (FRiP). Cells were filtered based on these metrics, as well as thresholds for minimum and maximum numbers of accessible peaks, determined through iterative QC visualization and empirical adjustment.

Normalization of the accessibility matrix was performed using term-frequency inverse document frequency (TF-IDF) transformation, followed by linear dimensionality reduction via singular value decomposition (SVD). Low-dimensional embedding was achieved using Uniform Manifold Approximation and Projection (UMAP), and cell clustering was performed using a shared nearest neighbor (SNN) graph-based method.

To quantify intra-sample heterogeneity and assess the robustness of clustering, we computed multiple metrics on the low-dimensional LSI embeddings and accessibility matrices. After clustering cells using a shared nearest neighbor (SNN) graph, silhouette widths were calculated based on the Euclidean distances in LSI space (dimensions 2–30) using the cluster R package (v2.1.6), providing a measure of separation and compactness for each condition. To evaluate local variability, we calculated the mean pairwise Euclidean distance in LSI space between each cell and all others, assigning the average distance as a per-cell heterogeneity score. In addition, we estimated information-theoretic diversity by computing Shannon entropy of accessibility counts per cell, where probabilities were defined as the normalized read counts across peaks. These metric distributions (silhouette width and entropy) were visualized using violin plots.

To validate the scATAC-seq findings, the data were processed as pseudo-bulk and analyzed alongside the bulk ATAC-seq data, following the same methodology described in the Methods section for bulk ATAC-seq analysis. Batch effect correction and dataset integration were performed using the ComBat_seq function from the sva (v3.44.0) package. Data normalization was carried out using the DESeq2 (v1.36.0) package. A PCA plot was generated to assess the similarity between the ATAC profiles of the bulk and pseudo-bulk (scATAC-seq) datasets. Downstream analysis, consistent with the bulk ATAC-seq pipeline, was conducted to identify differentially accessible peaks. A heatmap was produced to illustrate the similarity between samples from both data types.

## Bulk RNA sequencing of hiPSCs

200 HPCs (CD34+CD43+CD33+CD45+ and DAPI-) derived from parental and mutant hiPSCs were sorted at day 15 of EB differentiation directly in 8 μl lysis buffer (0,2% Triton and RNase-inhibitors (Thermo Scientific, #EO038SKB011)). For cDNA amplification Smart-seq2 method was used, generating full-length cDNA[78]. The quality of the cDNA was evaluated on the Agilent High-Sensitivity-DNA-chip by Agilent 2100 Bioanalyzer. The cDNA product was processed by Nextera Low-Input library kit for whole transcriptome sequencing on the HiSeq2000.

## Genome-wide gene expression analysis

The pipeline nf-core/rnaseq (v3.10.1) was used to process the raw RNA-seq data. This tool performs quality control (QC), trimming and alignment, and produces a gene expression matrix and extensive QC report. First, FastQC/MultiQC was used to perform quality checking and reporting of sequencing data. Then, reads were aligned to the hg38 reference genome using STAR (v2.7.10a) and gene expression was quantified with RSEM (v1.3.1). Downstream analysis was conducted using R (v4.2.0), specifically DESeq2 package (v1.36.0) was used. Differential expression analyses were carried out using a within-batch comparison. This approach effectively controlled technical variability and enabled robust identification of differentially expressed genes (DEGs) across all conditions. DEGs were identified using the Benjamini−Hochberg procedure to correct for multiple testing. Genes were considered significantly differentially expressed if they had an adjusted $p$-value $< 0.05$ and an absolute log2FC $> 0.5$. DEGs from the following contrasts (S_vs_P, A_vs_P, GSA_vs_P, GS_vs_P, GA_vs_P, and G_vs_P) were used for downstream functional enrichment analyses.

To integrate our bulk RNA-seq dataset with publicly available transcriptomic data, we utilized RNA-seq data from myelodysplastic neoplasm (MDS) patients published by Kotini et al.[23]. Starting from the same gene count matrices generated via the nf-core/rnaseq pipeline (v3.10.1), we downloaded and processed the Kotini dataset to ensure compatibility. To correct for batch effects between datasets, we employed the ComBat_seq function from the sva package (v3.44.0). Differential expression analyses were performed, after cohort-batch correction, across all conditions in the merged dataset using the same criteria described for our primary bulk RNA-seq analysis: genes with an adjusted $p$-value $< 0.05$ and an absolute log2FC $> 0.5$ were considered significantly differentially expressed. Heatmap visualization was generated to reveal striking similarities in expression profiles between MDS patient samples from the Kotini dataset and specific experimental groups in our study.

## Bulk ATACseq and Bulk RNAseq data integration

In order to explore the relationship between transcriptional and chromatin accessibility changes of cohort data, we integrated differential expression data from bulk RNA-seq with differential accessibility data from bulk ATAC-seq. For each condition, we compared the log2FC of DAPs with the log2FC of DEGs. DAPs and DEGs were defined as significant if they exhibited an absolute log2FC $> 0.5$ and a Benjamini−Hochberg adjusted $p$-value ($p$adj)$<0.05$. To assess the concordance between gene expression and chromatin accessibility, we assigned each DAP to its nearest gene and plotted the ATAC-seq DAP log2FC ($y$-axis) against the corresponding gene log2FC from RNA-seq DEGs ($x$-axis). The figures were generated using the ggplot2 R package (v3.5.1).

## Statistical analysis

Statistical analysis was performed with GraphPad-Prism 10 software. Data are shown as means with standard error of the mean (SEM). Pairwise comparisons between groups were performed using two-tailed unpaired Student's $t$ test, when normality was followed, otherwise, two-tailed Mann–Whitney test was used. For all analyses, the cutoff was set as $^*p < 0.05$; $^{***}p < 0.001$; $^{****}p < 0.0001$.

No statistical method was used to predetermine sample size. No data were excluded from the analyses. The experiments were not randomized.

## Ethical statement

The generation of hiPSC lines for this work was done under 2021 ISSCR Guidelines for Stem Cell Research and Clinical Translation and approved by the Ethics Committee of Clinical Research of the Center for Regenerative Medicine of Barcelona and the Catalan authority, Department of Health, approval numbers 4182562 and 525429.

## Reporting summary

Further information on research design is available in the Nature Portfolio Reporting Summary linked to this article.

## Data availability

The RNA-seq, ATAC-seq, and scATAC-seq data generated in this study have been deposited in the GEO (Gene Expression Omnibus) under accession code: GSE265824, GSE264140, and GSE300314, respectively. In addition to our ATAC-seq datasets, we used two publicly available ChIP-seq datasets: GATA2 ChIP-seq by Fujiwara et al. (GSE18829 [https://www.ncbi.nlm.nih.gov/geo/query/acc.cgi?acc=GSE18829]) and SETBP1 ChIP-seq by Piazza et al. (GSE86335 [https://www.ncbi.nlm.nih.gov/geo/query/acc.cgi?acc=GSE86335]). Furthermore, our bulk ATAC-seq profiles were compared with publicly available blood cell lineage ATAC-seq datasets reported by Corces et al. (GSE74246 [https://www.ncbi.nlm.nih.gov/geo/query/acc.cgi?acc=GSE74246]). Processed data generated in this study are provided in the Supplementary Tables. Source data are provided with this paper.

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

## Acknowledgements

We thank Dr. MA. Mulaw -Ulm University and Dr. J. Pozniak- KU Leuven for their valuable advice and suggestions regarding the scATAC-seq analysis. This work was supported by ERA PerMed GATA2-HuMo Funding Mechanism (Spain: Acció Instrumental de SLT011/18/00006 of the Department of Health of the Government of Catalonia), Fondazione Pisana per la Scienza ONLUS (FPS-G01/2018), Ministerio de Ciencia e Innovación, which is part of Agencia Estatal de Investigación (AEI), through the Retos Investigación grant, number PID2020-115591RB-I00 / https://doi.org/10.13039/501100011033, Fundació La Marató TV3 228/ C/2020, Award no. AC23_2/00040 by ISCIII through AES 2023 and within the European Joint Programme Rare Diseases framework, by Instituto de Salud Carlos III (ISCIII), "Programa FORTALECE del Ministerio de Ciencia e Innovación", through the project number FORT23/00032, and Finançat per el Departament de Recerca i Universitats de la Generalitat de Catalunya i l'AGAUR (expedient 2021 SGR 00888) to AG. Funding for this project was provided in part by an EHA Research Grant award granted by the European Hematology Association (KOG-202109-01162). J.P. and M.D. were supported by Fundació La Marató TV3 228/C/2020. Z.Q. and Z.L. are supported by funding from the Shenzhen Bay Laboratory Startup Fund. D.R.M. was supported by Deutsche José Carreras Leukämie-Stiftung, DJCLS 13R/2022. O.M.-B. was supported by 101029927-scGATA2track (H2020-MSCA-IF-2020) and he is currently supported by the Ramón y Cajal contract (RYC2021-032129-I) funded by AEI/European Social Fund UE. P.S. received the support of a fellowship from"la Caixa" Foundation (ID 100010434). The fellowship code is LCF/BQ/DI22/11940009. M.M.M. was supported by the Government of Andorra with the predoctoral fellowship, ATC030-AND/2022. The work in the LP laboratory was supported by "la Caixa" Foundation, LCF-PR-HR24-00150 and by PID2023-151556OB-I00 and CNS2024-154742 funded by MICIU/AEI/10.13039/501100011033 and, as appropriate, by "ESF Investing in your future", by "ESF+" or by "European Union NextGenerationEU/PRTR". MCF received funding from "la Caixa" Foundation (ID 100010434), under the agreement LCF/PR/HR20/52400014, PIXEL - TRANSCAN3-2021 (FAECC&ISCIII) and 2021 SGR 00888 - AGAUR 2022. The work in AB laboratory was funded by ERAPerMed-Departament de Salut, Generalitat de Catalunya (SLT011/18/00007) and the EJP RD-Instituto de Salud Carlos III (AC23_2/00014). This project has received funding from the European Union's Horizon 2020 research and innovation programme under the Marie Sklodowska-Curie grant agreement No 101029927. We thank Dr- Pablo Menendez and Dr. Clara Bueno group for technical support and expertise during experimental procedures. We thank the CERCA Programme/Generalitat de Catalunya for institutional support.

## Author contributions

O.M.-B. and A.G. designed the study and wrote the manuscript. J.P., D.R.-M., R.A., V.G.-H., M.M.M., J.C., f.d.G., Z.Q., A.I. and O.M.-B. performed cell biology experiments and data analysis. J.P., E.T., M.D., P.S., S.M.-V. and C.B.-B. performed the genomic studies and data analysis. L.P., Z.L., A.C., M.C.F., M.W.W., A.B., O.M.-B. and A.G. supervised data analysis. All authors contributed to the manuscript and provided final approval.

## Competing interests

The authors declare no competing interests.

## Additional information

[1]Hematopoietic Stem Cell Biology and Leukemogenesis, Regenerative Medicine Program, Bellvitge Institute for Biomedical Research (IDIBELL), L'Hospitalet de Llobregat, Spain. [2]Program for Advancing the Clinical Translation of Regenerative Medicine of Catalonia, P-CMR[C], Barcelona, Spain. [3]Programa de Doctorat de Medicina i Reserca Translacional, Facultat de Medicina i Ciències de la Salut, Universitat de Barcelona (UB), l'Hospitalet de Llobregat, Spain. [4]Stem Cell Aging Group, Regenerative Medicine Program, Institut d'Investigació Biomèdica de Bellvitge (IDIBELL), L'Hospitalet de Llobregat, Spain. [5]Centro de Investigación Biomédica en Red de Oncología (CIBERONC), Instituto de Salud Carlos III, Madrid, Spain. [6]Programa de Investigación en Cáncer, Hospital Del Mar Research Institute-Barcelona (HMRIB), Barcelona, Spain. [7]Endocrine Regulatory Genomics, Department of Medicine and Life Sciences, Universitat Pompeu Fabra (UPF), Barcelona, Spain. [8]Hospital Sant Joan de Deu Advanced Therapies Platform, SJD Pediatric Cancer Center Barcelona (PCCB) building, Barcelona, Spain. [9]Institute of Molecular Physiology, Shenzhen Bay Laboratory, Shenzhen, China. [10]Programa de Doctorat en Biomedicina, Universitat de Barcelona, Barcelona, Spain. [11]Josep Carreras Leukaemia Research Institute (IJC), Barcelona, Spain. [12]Centre for Genomic Regulation (CRG), Barcelona Institute of Science and Technology (BIST), Barcelona, Spain. [13]Universitat Pompeu Fabra (UPF), Barcelona, Spain. [14]Department of Hematology and Oncology, Institut de Recerca Sant Joan de Déu, Hospital Sant Joan de Déu, Barcelona, Spain. [15]Centro de Investigación Biomédica en Red de Enfermedades Raras (CIBERER), Instituto de Salud Carlos III, Madrid, Spain. [16]Center for Networked Biomedical Research on Bioengineering, Biomaterials and Nanomedicine (CIBER-BBN), Madrid, Spain. [17]The Catalan Institution for Research and Advanced Studies (ICREA), Barcelona, Spain. [18]Department of Hematology, St. Jude

Children's Research Hospital, Memphis, TN, USA. [19]Germans Trias i Pujol Health Science Research Institute (IGTP), Cancer Program, Badalona, Catalonia, Spain. [20]Department of Pathology and Experimental Therapeutics, Faculty of Medicine and Health Sciences, Barcelona University, Barcelona, Spain. [21]Fondazione Pisana Per la Scienza ONLUS (FPS), San Giuliano Terme, Italy. [22]Present address: Department of Physiology Development and Neuroscience, University of Cambridge, Cambridge, UK. ✉e-mail: omarin@igtp.cat; agiorgetti@idibell.cat

