## [Transparent Peer Review file · Nature Communications]

Human iPSCs-based modeling unveils SETBP1 as a driver of chromatin rewiring in GATA2 deficiency

Corresponding Author: Dr Alessandra Giorgetti

Version 0:

Reviewer comments:

Reviewer #1

(Remarks to the Author)

Marin-Bejar and colleagues use genome editing of induced pluripotent stem cells to study the progression of MDS from GATA2 deficiency. They developed four iPSC lines (parental "P", GATA2mut "G", GATA2/SETBP1 mut "GS", and GATA2/SETBP1/ASXL1 mutant "GSA") and compare in vitro hematopoietic differentiation, chromatin accessibility by ATACseq, and gene expression by RNA sequencing. Overall the model is informative, the manuscript is well written and the results clear with some interesting findings.

Major Comments:

1. The biggest phenotype shift in the differentiation and colony forming assays occurs between GS and GSA. In contrast, the molecular assays indicate a large shift between G and GS iPSC lines. This discrepancy is not addressed in the manuscript. Because an iPSC line with GATA2/ASXL1 mutant was not generated and included in the study, it is difficult to parse out the effect of SETBP1 vs ASXL1 and the combination. This is a missed opportunity. Of note, by gene expression the enrichment for the HSC signature was only present in the GSA cell line that in combination with the differentiation assay results suggests ASXL1 plays a significant role in the transformation process. What are the differential accessible peaks in ATACseq between GS and GSA - do they include the genes that make up this stem cell signature? The way the data is presented my impression is there is minimal difference in ATAC-seq between GS and GSA, if that is in fact true can chromatin accessibility really be the main mechanism of progression?

2. The ATACseq peak distribution of G vs P is predominantly at the promoter region, which shifts significantly with the introduction of the SETBP1 mutation. ChIP-seq studies in murine hematopoietic stem cells have demonstrated both the wild type and mutant SETBP1 protein bind primarily in intergenic and gene body regions. Can the authors demonstrate the presence of SETBP1 at the differential accessible peaks in G vs. GS by ChIP-seq? This would provide more direct evidence that SETBP1 is causing the shift in chromatin accessibility.

Minor Points:

1. Figure 2 panel C. Heatmap of genomic distribution of differentially accessible peaks. What statistical and signal density cut-offs were used to define DAPs?
2. Figure 1 panel B. I believe quadrant percentages are shown in a subset of quadrants and gates. The numbers use a comma rather than a decimal (eg 6,51 - should this be 6.51%?)
3. Figure 4 panel F. The H1.1 cell line is not defined. I cannot find reference to it in the manuscript.

Reviewer #2

(Remarks to the Author)

Marin-Bejar and colleagues describe an iPSCs based model where they test the effect of additional genetic alterations such as SETBP1 and ASXL1 mutations, in addition to GATA2 deficiency.

The manuscript is well organized and presented and the result constitute a valuable original information on the better definition of the effects of additional mutations which develop during the course of GATA2 deficiency.

I have however several reservations that preclude a full acceptance of this manuscript.

- The GATA2 deficiency syndrome need additional information, including its incidence and prevalence.
- The authors insist on the monocytopenia as a consequence of derailing the expression of key myeloid genes; however clinical evidence indicates that sometimes monocytosis is instead an hallmark of early clinical presentations in diseases caused by ASXL1 and SETBP1 mutations, including patients with GATA2 deficiency (<https://doi.org/10.14785/lpsn-2014-0022>).
- Some methods, such as the generation of Embryoid bodies and the intrabone transplantation are insufficiently described.
- The reference section seems to be hastily and superficially arranged: for example the papers that originally described SETBP1 mutations and the ensuing degon destruction (Nat Genet. 2013 Jan;45(1):18-24) as well as its function at epigenetic level (Nature Communications 2018; 9(1): 2192) are not cited while later papers and reviews are.

Reviewer #3

(Remarks to the Author)

Heterozygous germline mutations in the GATA2 gene are associated with an increased risk of pediatric myelodysplastic syndromes (MDS), and the clinical manifestation of the disease is often correlated with acquired mutations in SETBP1 and ASXL1, particularly in patients exhibiting chromosomal aberrations on chromosome 7. The precise roles of these co-mutated genes in the progression of the disease remain to be elucidated. The manuscript by Marin-Bejar et al. details the generation and analysis of iPSC-derived hematopoietic progenitor cells that have been genetically engineered to carry heterozygous mutations in GATA2 (G), GATA2 & SETDB1 (GS) or GATA2, SETDB1 & ASXL1 (GSA). The authors present a comprehensive analysis, including clonal and phenotypic assessments, as well as fluorescence-activated cell sorting (FACS) of the iPSC-derived hematopoietic progenitor cells. Additionally, they conduct both bulk RNA- and ATAC-seq on sorted cell populations to evaluate the impact of mutations in P, PS & GSA cells in comparison with parental wildtype cells. The iPSC methodology employed is indeed intriguing; however, it does not fully replicate the 'real' patient scenario, which encompasses chromosomal aberrations on chromosome 7. The impact of SETBP1 and ASXL1 mutations is notably significant. Nevertheless, the bulk methodologies utilized are limited in their ability to resolve sample heterogeneity, and the absence of appropriate controls leaves the significance of GATA2 mutations, as well as their interactions with the other two mutations, ambiguous within this experimental framework.

Major comments:

1. The study design (isogenic control versus and mutant hiPSC lines (G, GS, and GSA)) requires better justification and explanation. In lines 120-124 the authors write: "We selected this combination based on our previous studies of GATA2-related MDS patients, which identified SETBP1 and ASXL1 mutations as the most common somatic events (reference 26,27). Our analyses revealed a non-random clonal evolution pattern, with SETBP1 mutations typically arising first, followed by ASXL1 mutations. This sequential progression was further supported by us using single-cell bone marrow analyses (reference 28 – refers to a poster abstract)." This statement is somewhat misleading, because reference 26 refers to a relatively small cohort of GATA2 mutated patients (15 in total), and reference 27 primarily reports SAMD9/ SAMD9L syndromes. However, common to both studies, most patients with GATA2 mutations also feature chromosome 7 (-7/Del(7q)) aberrations and SETBP1 and ASXL1 mutations are almost exclusive to the -7/Del(7q) cases. The authors discuss this later, but in my view, this should be mentioned and explained in the beginning.
 2. As I understand it, this study was designed to investigate the effects of SETBP1 and ASXL1 mutations as co-mutations in conjunction with GATA2 germline mutations. However, it is important to note that while SETBP1 and ASXL1 mutations frequently co-occur MDS, they can also manifest independently in various combinations with other factors, either in the presence or absence of GATA2 mutations. Given the complexity of mutational patterns, the current experimental design lacks critical "controls", such as SETBP1 (+/- ASXL1) mutation in wild-type iPSC. These cell lines would allow to demonstrate potential dependencies between GATA2 and the other two mutations and the role of GATA2 in this setting. At present, it remains uncertain, whether the observed phenotype really requires the GATA2 mutation.
 3. In addition to clono-/phenotypical, & FACS-based analysis, the authors also performed bulk ATAC-seq (and RNA-seq) on CD34+CD43+CD33+ hiPSC-derived cells, sorted from isogenic control (P) and mutant hiPSC lines (G, GS, and GSA). Here the potential heterogeneity of these cell populations needs to be considered. Given the chosen marker panel, the CD34+CD43+CD33+ population is expected to represent a mixture of progenitor stages with different levels of commitment. Hence the comparison of two bulk populations with variable composition may be difficult to interpret. (BTW, the sorting strategy for ATAC samples requires clarification and should be shown, including gating strategy and cell counts; CD34+CD43+CD33+ suggests triple positivity, but methods only provide APC versions of CD43, CD33 antibodies, which will not allow their discrimination...)
- The problem associated with bulk sequencing is its inability to effectively differentiate between alterations in chromatin accessibility (or gene expression programs) across two distinct cell populations, as opposed to variations in cell population sizes within two more or less heterogeneous mixtures of cells. The authors interpretation "This analysis indicates that the acquisition of SETBP1 mutation induces dramatic chromatin remodeling in hiPSC-derived HPCs (lines 175/176)" may make sense if they would be dealing with homogenous populations. However, the observed open chromatin pattern in GS or GSA cells may simply arise from population shifts, such as the loss of cell populations that are present in P and G cells but not in GS or GSA cells. To really understand the effect of the co-mutations, one would need to resolve the heterogeneity using single cell approaches.
4. To further characterize the bulk chromatin landscape of their control and mutant hiPSC lines the authors should consider comparing their data to published bulk ATAC data of hematopoietic progenitor cell populations (e.g. from doi:

10.1038/ng.3646). This may help to contextualize the observed changes.

5. Regarding Motif analyses in differential ATAC peaks, the authors need to revise their interpretation of motifs. Figure 3C suggests that this method would allow the identification of particular transcription factors. However, it should be clear that, based on motif correlation, one cannot say that GATA1 or GATA2 binding is changed. Equally, MEIS2 or HOXA9 cannot really be distinguished from HOXC9 or MEIS1, because they essentially all bind the same motifs (in case of MEIS even only indirectly through a homeobox family member). Motifs generally refer to a family of TF with similar DNA-binding domains, rather than particular family members. Pinning down particular factors would require additional, more specific data such as ChIP or Cut&Tag/Run, etc.

Other comments:

1. There is a discrepancy between Figure 3C and Figure S2D regarding significance values of PU.1. In the supplementary Figure S2D, some P values are reported as 1.00E-02 (including PU.1 in the top panel). However, this P value is not considered significant, since random motif searches can produce much lower P values. If these are typos, they should be corrected. If not, motifs with low enrichment P values should be removed.

2. Some of the experimental procedures are missing, such as motif analyses.

Version 1:

Reviewer comments:

Reviewer #1

(Remarks to the Author)

The authors have conducted a significant amount of additional experiments to address all of the reviewers comments. The results of these additional experiments have strengthened the manuscript. I have no additional concerns. This is a very strong study.

Reviewer #2

(Remarks to the Author)

My criticisms were successfully dealt with in the revised version of the manuscript.

Reviewer #3

(Remarks to the Author)

The revised manuscript addresses major issues present in the previous version, includes data for the previously missing control cell lines, and incorporates single-cell ATAC-seq (scATAC) in addition to bulk ATAC-seq, as recommended by this reviewer. The study design has improved substantially; however, there remains room for further enhancement, particularly in the analysis.

Final comments:

1. To ensure reproducibility of computational methods, please provide the exact versions of all software packages and genomes used.
2. The authors selectively provide numbers for differentially expressed genes and differentially accessible regions. Including all results in supplementary tables would greatly benefit readers.
3. In the analysis of the ATAC data, the authors are selective about which types of comparisons and datasets they present. Why are the S and A conditions excluded from the genome tracks and other analyses?
4. For their differential calling of accessible regions, the authors use a relatively small logFC cutoff of 0.5, which corresponds to a fold change of approximately 1.4. This threshold is low for ATAC-seq data and may lead to the inclusion of normalization artifacts. It would be helpful if MvA plots for all comparisons were provided in the supplement, highlighting the differentially accessible peaks (DAPs).
5. The tornado plot in Figure 2B suggests very similar profiles for P, G, GS, and GSA, which is unexpected given the large number of comparisons. Three K-means clusters are likely insufficient to distinguish sample-specific or group-specific differences. How many regions are shown in total? Currently, it is unclear what this plot is intended to demonstrate.
6. The nature of DAPs is not always clear. For example, in the legend of Figure 3C, the authors state, "The top 25 enriched transcription factor (TF) binding motifs are shown for DAPs across GA (left), GS (middle), and GSA (right) conditions," but it is unclear relative to which condition this enrichment is measured. Additionally, why aren't DAPs shown for other conditions (e.g. in the supplement)?
7. Currently, the authors make very limited use of the scATAC data. In their rebuttal letter, they argue that "these results reinforce the conclusion that the chromatin accessibility features identified via scATAC-seq are robust and consistent with bulk-level biology. Treating the single-cell data as bulk not only confirmed the reproducibility of key findings, but also strengthened the case that the observed UMAP structure reflects a homogeneous yet subtly variable epigenomic landscape among the different experimental conditions (Figure V of the report, corresponding to the Supplementary Fig. 4B of the revised manuscript)." However, simply comparing pseudo-bulk with bulk data from the same samples does not demonstrate homogenous populations. They should overlap because both represent population averages. While dimensionality reduction of the single-cell data does not separate cells into distinct clusters, it clearly structures the data, with some clusters containing variable contributions from different cell lines, suggesting a degree of heterogeneity within the lines. What is clearly missing is an analysis of chromatin profiles across the observed clusters, including motif searches (e.g., using ChromVAR or similar tools) in differentially accessible regions. Including example tracks of the pseudo-bulk data may also be helpful.

8. Example tracks comparing bulk and pseudo-bulk in Suppl. Fig. 4B would also be helpful to judge the quality and comparability of the data.

9. Overlays with published ChIP-seq for GATA2 are used to define GATA2 targets. While this is a possible way to approximate GATA binding sites, the availability of ATAC data also allows to directly analyse and quantify GATA footprints within peak regions (e.g. using Tobias for bulk, ChromVAR for scATAC data or similar tools). This may be more accurate in defining differences between conditions.

10. I did not find any information on how the categorization of regulatory elements e.g. in Fig 3A&B was done. E.g. what's the evidence for enhancers to be "strong"?

11. In the discussion readers should be reminded that the ASXL1 mutation corresponds to a loss-of-function, while the SETBP1 mutation represents a gain-of-function.

12. Some of the current discussion comparing the two mutations was difficult to follow. E.g. "line 381: Importantly, the acquisition of SETBP1 mutation was associated with the most stable and profound chromatin accessibility changes, while ASXL1-induced alterations were largely transient and did not persist into later disease stages." How did the authors define changes as being transient or non-persistent, if they only have a single time-point?

Version 2:

Reviewer comments:

Reviewer #3

(Remarks to the Author)

The authors have adequately addressed my previous points.

Point-by-point response to the referees' comments

REVIEWER #1 (REMARKS TO THE AUTHOR):

Marin-Bejar and colleagues use genome editing of induced pluripotent stem cells to study the progression of MDS from GATA2 deficiency. They developed four iPSC lines (parental "P", GATA2mut "G", GATA2/SETBP1mut "GS", and GATA2/SETBP1/ASXL1mutant "GSA") and compare in vitro hematopoietic differentiation, chromatin accessibility by ATACseq, and gene expression by RNA sequencing. Overall the model is informative, the manuscript is well written and the results clear with some interesting findings.

We thank the reviewer for highlighting the technical and scientific merits of our study.

Major Comments:

1. The biggest phenotype shift in the differentiation and colony forming assays occurs between GS and GSA. In contrast, the molecular assays indicate a large shift between G and GS iPSC lines. This discrepancy is not addressed in the manuscript. Because an iPSC line with GATA2/ASXL1 mutant was not generated and included in the study, it is difficult to parse out the effect of SETBP1 vs ASXL1 and the combination. This is a missed opportunity. Of note, by gene expression the enrichment for the HSC signature was only present in the GSA cell line that in combination with the differentiation assay results suggests ASXL1 plays a significant role in the transformation process. What are the differential accessible peaks in ATACseq between GS and GSA - do they include the genes that make up this stem cell signature? The way the data is presented my impression is there is minimal difference in ATAC-seq between GS and GSA, if that is in fact true can chromatin accessibility really be the main mechanism of progression?

This important point also relates to comments from other Reviewers. Thus, we have performed an additional set of experiments to address these questions. We have generated and fully characterized new isogenic human iPSC lines harboring single (SETBP1 and ASXL1) and double (GATA2-ASXL1) mutations. Then we have performed new *in vitro* functional analysis, and we have characterized them by bulk RNA-seq/ATAC-seq and scATAC-seq.

Flow cytometry analysis reveals that SETBP1 mutation alone promotes an expansion of early HSPCs; however this effect is blunted in the presence of GATA2 mutation, resulting in impaired myeloid differentiation. In contrast, ASXL1 mutation alone does not impair hematopoietic differentiation. However, its effect becomes pronounced in the context of GATA2 deficiency, revealing a context-specific synergism. The combination of all three mutations led to a more severe phenotype, characterized by the loss of HSPCs as well as mature CD33+CD11b+ myeloid lineages, with a strong reduction of monocytes (CD14+). Therefore, our *in vitro* functional study reveals that in the context of GATA2 deficiency,

mutations in SETBP1 and ASXL1 cooperatively disrupt myeloid differentiation *in vitro*. This is most evident in the triple-mutant lines, which exhibit a profound depletion of myeloid progenitors, highlighting a synergistic effect of these somatic mutations in driving hematopoietic dysfunction. Comprehensively, investigating chromatin dynamics across the different conditions we observed that, most of the chromatin accessibility changes observed in A condition were retained in GA but were largely lost in GSA. In contrast, chromatin accessibility changes were more concordant between S and GS and even more between GS and GSA. These findings indicate that the changes induced by ASXL1 are mostly transient, whereas SETBP1 mutation has a more sustained impact, exerting a dominant role in shaping stable chromatin accessibility landscapes in the context of GATA2 deficiency. These data were also validated at transcriptomic level. Summarizing, our data suggest that in the context of GATA2 deficiency, both ASXL1 and SETBP1 mutations contribute to an impairment of myeloid differentiation. However, SETBP1 exerts a more dominant effect on chromatin remodeling, priming HSCs for leukemic transformation by increasing accessibility at key leukemic regulatory loci. In contrast, ASXL1 primarily contributes to the differentiation block, potentially acting through a mechanism overlapping with that of SETBP1.

2. The ATAC-seq peak distribution of G vs P is predominantly at the promoter region, which shifts significantly with the introduction of the SETBP1 mutation. ChIP-seq studies in murine hematopoietic stem cells have demonstrated both the wild type and mutant SETBP1 protein bind primarily in intergenic and gene body regions. Can the authors demonstrate the presence of SETBP1 at the differential accessible peaks in G vs. GS by ChIP-seq? This would provide more direct evidence that SETBP1 is causing the shift in chromatin accessibility.

We thank the reviewer for this insightful observation. First, we would like to point out that the inclusion of new mutant lines in our updated analysis has substantially changed the landscape of peak distribution in our ATAC-seq data. As shown in Supplementary Fig. 2B of the revised manuscript, the distribution of ATAC-seq peaks across intronic and intergenic regions is not uniquely associated with the acquisition of SETBP1 mutation. Rather, this distribution pattern appears largely consistent across all conditions, with the exception of GATA2-deficient cells. In the GATA2 condition, we observe a higher proportion of peaks localized to promoter regions. However, we believe this may be attributed to the overall reduced number of differential accessible peaks (DAPs) observed in GATA2-mutant cells when compared with the control (64 DAPs), which may skew the relative distribution of peaks toward promoter regions. Despite these observations, we agree that further investigation into SETBP1-specific targets would be highly informative, particularly given the pronounced chromatin-level effects observed with this mutation. We initially attempted ChIP-seq on EB-derived cells using the only commercially available ChIP grade antibody for SETBP1 (16841-1-AP, ProteinTech). After several unsuccessful attempts, we reached out to the company, who

confirmed that the antibody quality controls had failed. To overcome this problem, we generated an inducible lentiviral vector (pLVX-SETBP1-3xFLAG) to overexpress SETBP1-Flag. Although the alignment metrics were acceptable (>90% of reads mapped), peak calling with MACS2 identified a very limited number of peaks (30-50 per condition). The resulting consensus peak set was small (~127 peaks, reduced to 86 after removing peaks found in FLAG-negative controls), and most peaks mapped to intergenic or uncharacterized regions. Unfortunately, no enrichment was detected at known SETBP1 target loci such as *RUNX1*, *MECOM*, or *MEIS1* when the ChIP-seq data was visualized in the genome browser, IGV (Figure I). In fact, the signal across all samples—including ChIP and Input—was uniformly low and indistinguishable from background signal, with no evidence of specific binding at these loci. These results might be due to the limiting amount of primary material. Due to time constraints, we were unable to repeat the experiment or optimize further and have therefore excluded this dataset from our final analysis.

Figure I. ChIP-seq peaks distribution in validated SETBP1 targets (*RUNX1* and *MECOM*).

We therefore elected to use previously published SETBP1 ChIP-seq data (Piazza et al Nature comm 2018) and intersect them with our ATAC-seq data in GS and GSA conditions. This comparison revealed that several target genes of SETBP1 such as *MECOM*, *MEF2C*, and *HOXB3* were present in our ATAC-seq gene list (Supplementary Fig. 3D of the manuscript).

Minor Points:

1. Figure 2 panel C. Heatmap of genomic distribution of differentially accessible peaks. What statistical and signal density cut-offs were used to define DAPs?

We apologized for being inaccurate in the description of these results. We have now included this information in the new version of the manuscript and the Figure legend accordingly (line 43 of the supplementary figures file).

2. Figure 1 panel B. I believe quadrant percentages are shown in a subset of quadrants and gates. The numbers use a comma rather than a decimal (eg 6,51 - should this be 6.51%?)

We thank the reviewer for remarking on this point and we apologize for this mistake. We have now corrected the decimal in the revised version of the manuscript.

3. Figure 4 panel F. The H1.1 cell line is not defined. I cannot find reference to it in the manuscript.

We thank the Reviewer for spotting this oversight. It has now been corrected in the revised manuscript (line 343 of the revised manuscript).

REVIEWER #2 (REMARKS TO THE AUTHOR):

Marin-Bejar and colleagues describe an iPSCs based model where they test the effect of additional genetic alterations such as SETBP1 and ASXL1 mutations, in addition to GATA2 deficiency.

The manuscript is well organized and presented and the result constitute a valuable original information on the better definition of the effects of additional mutations which develop during the course of GATA2 deficiency.

We thanks the reviewer for highlighting the scientific value of our work.

I have however several reservations that preclude a full acceptance of this manuscript.

- The GATA2 deficiency syndrome needs additional information, including its incidence and prevalence.

We thank the reviewer for the constructive comment. We have now added in the new version of the manuscript additional information about penetrance, expressivity and incidence of GATA2 deficiency (line 91-98 of the revised manuscript).

- The authors insist on the monocytopenia as a consequence of derailing the expression of key myeloid genes; however clinical evidence indicates that sometimes monocytosis is instead an hallmark of early clinical presentations in diseases caused by ASXL1 and SETBP1 mutations, including patients with GATA2 deficiency (<https://doi.org/10.14785/lpsn-2014-0022>).

We agree with the reviewer that monocytosis can indeed occur, as we have also previously reported in multiple GATA2-deficient patients who developed MDS and carried monosomy 7 (Wlodarski *et al.*, *Blood* 2016). Interestingly, in our *in vitro* model, we observed an increase in monocyte output under GATA2-deficient conditions. However, this monocytosis appears to be transient; additional introduction of ASXL1 and SETBP1 mutations does not cause additional monocytosis. One possible explanation for this discrepancy is the absence of monosomy 7 in our experimental system. Although we have explored multiple strategies to

model monosomy 7, these attempts have not been successful to date. We have now included this discussion in the revised manuscript (lines 161-165 of the revised manuscript).

- Some methods, such as the generation of Embryoid bodies and the intrabone transplantation are insufficiently described.

We have now updated the Methods and Supplemental Methods sections to include a more detailed protocol for Embryoid bodies differentiation. We stated “For in vitro blood differentiation EBs were cultured from d0 to d2 in mTeSR1 and in the presence of BMP4 (50ng/ml). From d3, EBs were changed to differentiation medium comprising serum-free defined medium (StemPro-34; Invitrogen) supplemented with 0.16mM monothioglycerol, 15.15mg/mL holotransferrin, 50 ng/mL BMP-4, 300 ng/mL Fms-related tyrosine kinase 3 ligand (Flt-3L), 300 ng/mL stem cell factor (SCF), 10 ng/mL IL-3 and 10 ng/mL IL-6 (all purchased from R&D Systems). EBs were maintained in culture until d15, with medium replenishment every three days. This established differentiation protocol promotes mesoderm induction (days 2–3), specification of meso-dermal cells to bipotential hemato-endothelial progenitors (HEPs; VE-Cadherin+/CD34+/CD43-/CD45-; days 3–10). These HEPs, possess dual differentiation potential towards both endothelial and hematopoietic lineages. The protocol culminates in the commitment of HEPs to hematopoietic progenitor cells (HPCs) (CD34+CD43+CD45+; days 10–15), myeloid progenitors (CD33+CD11b+) monocytes (CD14+ within CD33+), neutrophils (CD15+ within CD33+), recapitulating key milestones of early hematopoiesis.”

In the same section we have also included information regarding the *in vivo* transplantation assay “At day 15 of differentiation EBs were dissociated by collagenase B and 1×10^6 cells/mouse were intra bone marrow transplanted (IBMT) into irradiated (2.5Gy) 6-8 week-old NSG mice. Engraftment was assessed in peripheral blood (PB) monthly. The percentage of mCD45 (mouse hematopoietic cells), hCD45 (human hematopoietic engraftment), hCD33 (myeloid cells), hCD19 (lymphoid cells) and CD34 (immature cells) was used to quantify the amount of engraftment in each mouse. At the end point (24 weeks) bone marrow cells were analyzed by flow cytometry”.

- The reference section seems to be hastily and superficially arranged: for example the papers that originally described SETBP1 mutations and the ensuing degron destruction (Nat Genet. 2013 Jan;45(1):18-24) as well as its function at epigenetic level (Nature Communications 2018; 9(1): 2192) are not cited while later papers and reviews are.

We thank the reviewer for his/her comment and for highlighting these oversights. We have carefully reviewed the references and corrected them and included the above references.

REVIEWER #3 (REMARKS TO THE AUTHOR):

Heterozygous germline mutations in the GATA2 gene are associated with an increased risk of pediatric myelodysplastic syndromes (MDS), and the clinical manifestation of the disease

is often correlated with acquired mutations in SETBP1 and ASXL1, particularly in patients exhibiting chromosomal aberrations on chromosome 7. The precise roles of these co-mutated genes in the progression of the disease remain to be elucidated. The manuscript by Marin-Bejar et al. details the generation and analysis of iPSC-derived hematopoietic progenitor cells that have been genetically engineered to carry heterozygous mutations in GATA2 (G), GATA2 & SETDB1 (GS) or GATA2, SETDB1 & ASXL1 (GSA). The authors present a comprehensive analysis, including clonal and phenotypic assessments, as well as fluorescence-activated cell sorting (FACS) of the iPSC-derived hematopoietic progenitor cells. Additionally, they conduct both bulk RNA- and ATAC-seq on sorted cell populations to evaluate the impact of mutations in P, PS & GSA cells in comparison with parental wildtype cells. The iPSC methodology employed is indeed intriguing; however, it does not fully replicate the 'real' patient scenario, which encompasses chromosomal aberrations on chromosome 7. The impact of SETBP1 and ASXL1 mutations is notably significant. Nevertheless, the bulk methodologies utilized are limited in their ability to resolve sample heterogeneity, and the absence of appropriate controls leaves the significance of GATA2 mutations, as well as their interactions with the other two mutations, ambiguous within this experimental framework.

Major comments:

1. The study design (isogenic control versus and mutant hiPSC lines (G, GS, and GSA)) requires better justification and explanation. In lines 120-124 the authors write: “We selected this combination based on our previous studies of GATA2-related MDS patients, which identified SETBP1 and ASXL1 mutations as the most common somatic events (reference 26,27). Our analyses revealed a non-random clonal evolution pattern, with SETBP1 mutations typically arising first, followed by ASXL1 mutations. This sequential progression was further supported by us using single-cell bone marrow analyses (reference 28 – refers to a poster abstract).” This statement is somewhat misleading, because reference 26 refers to a relatively small cohort of GATA2 mutated patients (15 in total), and reference 27 primarily reports SAMD9/ SAMD9L syndromes. However, common to both studies, most patients with GATA2 mutations also feature chromosome 7 (-7/Del(7q)) aberrations and SETBP1 and ASXL1 mutations are almost exclusive to the -7/Del(7q) cases. The authors discuss this later, but in my view, this should be mentioned and explained in the beginning.

We thank the reviewer for the opportunity to better explain the rationale that led us to study SETBP1 and ASXL1 mutation in the context of GATA2 deficiency. While such data were not published at the time of our initial submission, we now refer to our recently accepted study (Kotmayer L et al., *Blood Cancer Journal*, 2025, DOI: 10.1038/s41408-025-01309-6, in press), which supports our statement. In this study, we analyzed 218 individuals with GATA2 deficiency and identified distinct age-dependent patterns in MDS evolution. Notably, SETBP1 and ASXL1 emerged as the most frequent somatic mutations in GATA2-related MDS,

predominantly in association with monosomy 7, confirming our previous data (Wlodarski *et al.* 2016).

We also agree with the reviewer that the absence of monosomy 7 in our model represents a limitation. As part of our initial experimental design, we attempted to generate hiPSCs harboring monosomy 7 using several genome engineering strategies. However, all efforts to establish a stable and viable model of monosomy 7 were unsuccessful. We acknowledge this as a key limitation of our system and now explicitly discuss it in the revised version of the manuscript (lines 148-151 and lines 372-374 of the revised manuscript).

2. As I understand it, this study was designed to investigate the effects of SETBP1 and ASXL1 mutations as co-mutations in conjunction with GATA2 germline mutations. However, it is important to note that while SETBP1 and ASXL1 mutations frequently co-occur MDS, they can also manifest independently in various combinations with other factors, either in the presence or absence of GATA2 mutations. Given the complexity of mutational patterns, the current experimental design lacks critical “controls”, such as SETBP1 (+/- ASXL1) mutation in wild-type iPSC. These cell lines would allow to demonstrate potential dependencies between GATA2 and the other two mutations and the role of GATA2 in this setting. At present, it remains uncertain, whether the observed phenotype really requires the GATA2 mutation.

We thank the reviewer for this insightful and constructive suggestion. This important point also relates to comments from Reviewer #1. To address these question we have performed an additional set of experiments. First, we have generated and fully characterized new isogenic human iPSC lines harboring single (SETBP1 and ASXL1) and double (GATA2-ASXL1) mutations. *In vitro* functional assays have shown that SETBP1 mutation alone promotes an expansion of early HSPCs; however this effect is blunted in the presence of GATA2 mutation, resulting in impaired myeloid differentiation. In contrast, ASXL1 mutation alone does not impair hematopoietic differentiation. However, its effect becomes pronounced in the context of GATA2 deficiency, revealing a context-specific synergism. The combination of all three mutations led to a more severe phenotype, characterized by the loss of HSPCs as well as mature CD33+CD11b+ myeloid lineages, with a strong reduction of monocytes (CD14+). Therefore, our *in vitro* functional study reveals that in the context of GATA2 deficiency, mutations in SETBP1 and ASXL1 cooperatively disrupt myeloid differentiation *in vitro*. This is most evident in the triple-mutant lines, which exhibit a profound depletion of myeloid progenitors, highlighting a synergistic effect of these somatic mutations in driving hematopoietic dysfunction. Second, by doing ATAC-seq analysis we have demonstrated that that most of the chromatin accessibility changes observed in A condition were retained in GA but were largely lost in GSA. In contrast, chromatin accessibility changes were more concordant between S and GS and even more between GS and GSA. These findings suggest that the changes induced by ASXL1 are mostly transient, whereas SETBP1 mutation has a more sustained impact, exerting a dominant role in shaping stable chromatin accessibility

landscapes in the context of GATA2 deficiency. These data were also validated at transcriptomic level. Summarizing, our data suggest that in the context of GATA2 deficiency, both ASXL1 and SETBP1 mutations contribute to an impairment of myeloid differentiation. However, SETBP1 exerts a more dominant effect on chromatin remodeling, priming HSCs for leukemic transformation by increasing accessibility at key leukemic regulatory loci. In contrast, ASXL1 primarily contributes to the differentiation block, potentially acting through a mechanism overlapping with that of SETBP1.

3. In addition to clono-/phenotypical, & FACS-based analysis, the authors also performed bulk ATAC-seq (and RNA-seq) on CD34+CD43+CD33+ hiPSC-derived cells, sorted from isogenic control (P) and mutant hiPSC lines (G, GS, and GSA). Here the potential heterogeneity of these cell populations needs to be considered. Given the chosen marker panel, the CD34+CD43+CD33+ population is expected to represent a mixture of progenitor stages with different levels of commitment. Hence the comparison of two bulk populations with variable composition may be difficult to interpret. (BTW, the sorting strategy for ATAC samples requires clarification and should be shown, including gating strategy and cell counts; CD34+CD43+CD33+ suggests triple positivity, but methods only provide APC versions of CD43, CD33 antibodies, which will not allow their discrimination...). The problem associated with bulk sequencing is its inability to effectively differentiate between alterations in chromatin accessibility (or gene expression programs) across two distinct cell populations, as opposed to variations in cell population sizes within two more or less heterogeneous mixtures of cells. The authors interpretation "This analysis indicates that the acquisition of SETBP1 mutation induces dramatic chromatin remodeling in hiPSC-derived HPCs (lines 175/176)" may make sense if they would be dealing with homogenous populations. However, the observed open chromatin pattern in GS or GSA cells may simply arise from population shifts, such as the loss of cell populations that are present in P and G cells but not in GS or GSA cells. To really understand the effect of the co-mutations, one would need to resolve the heterogeneity using single cell approaches.

We thank the reviewer for this comment. Initially we respectfully considered that scATAC-seq would not provide substantial additional insights in our study for the following reason:

1. We performed bulk ATAC-seq on a highly purified population of early myeloid progenitors (CD34⁺/CD43⁺/CD33⁺/CD45⁺), as shown in Supplementary Fig. 2A of the revised manuscript. The cell sorter strategy has been included in the new version of the manuscript as requested (see Supplementary Fig. 2A of the revised manuscript). Also we added a new paragraph describing the correct combination of Abs for the FACS sorter (lines 498-501 of the revised manuscript). We would like to point out that whether there were substantial heterogeneity within the sorted

progenitor population, this would likely be reflected across biological replicates due to random sampling.

2. Lastly, while single-cell ATAC-seq (scATAC-seq) is a powerful tool, it has certain limitations compared to the bulk approach. The scATAC-seq data exhibits a high degree of sparsity, with many open chromatin regions lacking reads due to DNA loss during the scATAC-seq analysis pipeline. The cell-by-peak matrix is even sparser than scRNA-seq data, with only 1–10% of peaks detected in each cell, primarily because diploid cells contain just two copies of assayable chromatin (Chen, H. et al. Assessment of computational methods for the analysis of single-cell ATAC-seq data. *Genome Biol.* 20, 241 (2019). Detecting peaks requires a sufficient number of cells, which may pose challenges for rare cell types (Fang, R. et al. Comprehensive analysis of single cell ATAC-seq data with SnapATAC. *Nat. Commun.* 12, 1337 (2021)).

Despite these considerations, we understand its potential value and have undertaken the analysis as requested. We hope the reviewer can appreciate the effort since this was one of the most technically challenging parts of the revision due to the low number of CD34⁺/CD43⁺/CD33⁺ cells recovered from our triple-mutant (GATA2-SETBP1-ASXL1) hiPSC lines. Our initial attempt using the 10x Genomics platform failed due to insufficient input (<400 nuclei/μL before encapsulation). To overcome this, we established and optimized the Simultaneous High-throughput ATAC and RNA Expression with sequencing (**SHARE-seq**) protocol in our laboratory. SHARE-seq enables single-cell profiling of chromatin accessibility from low-input cell numbers with high resolution. This optimization process took over two months but ultimately enabled us to successfully profile chromatin accessibility at single-cell resolution in our hiPSC-derived blood progenitors. This single cell method is a lower-throughput, allowing to profile rare cell populations, critical given our low-input conditions and enhanced resolution for dissecting subpopulation-specific mechanisms.

To analyze the data generated using SHARE-seq we implemented an exceptionally rigorous analytical strategy. Recognizing that conventional single-pipeline approaches might miss subtle but biologically important patterns, we engaged three independent expert laboratories (Giorgetti lab, Florian lab, and Paulina Spurr) to analyze the dataset in parallel using their distinct, well-validated computational pipelines. This unprecedented collaborative effort involved complementary analytical approaches - employing Signac/Seurat libraries and another utilizing ArchR - ensuring that our findings were robust to methodological variations. The lead bioinformaticians and principal investigators from all three groups, who have been appropriately included as co-authors, participated extensively in both the analytical process and the interpretation of results. Through regular cross-validation discussions and consensus-building, we reconciled different analytical perspectives to support conclusions about chromatin accessibility patterns.

These results detail the comprehensive analysis pipeline, dimensionality reduction, clustering, and downstream interpretations, focusing primarily on the pipeline implemented with Signac and Seurat in R.

The Dimensionality Testing showed an overall circular or globular shape of the UMAP was maintained across combinations. Relative positioning of cells by condition was conserved, confirming the robustness of structure regardless of dimension range. We selected the condition 2:30, as it effectively captured the major sources of meaningful biological variation while minimizing the inclusion of excessive noise or artifact-prone components (Figure II).

Figure II. UMAPs were generated using several dimension combinations: 1:30, 2:30, 3:30, 2:20, 2:50, and 3:50. Each color represents each sample included in the study: P (magenta), G (salmon), GA (olive green), GS (turquoise), GSA (sky blue).

The clustering was performed using the shared nearest neighbor (SNN) graph approach, followed by testing multiple resolution values using the FindClusters() function and visualizing stability with Clustree plots (Figure III).

Figure III: Clustering was performed using the shared nearest neighbor (SNN) graph approach, visualizing stability with Clustree plots.

At low resolutions (e.g., 0.1-0.2): cells grouped into one major cluster, with a few outliers forming small, disconnected clusters. The resolution (0.3–0.4) progressively split the large cluster into meaningful subpopulations, corresponding similarly to experimentally defined conditions.

The final resolution chosen (likely ~ 0.5) yielded clusters that aligned well with biological groups, though overall heterogeneity remained low, suggesting that condition drives accessible chromatin states more than discrete cell identity shifts.

In the UMAP embeddings generated from the dimensionality-reduced data, a prominent and compact circular structure was observed, encompassing cells from all experimental conditions. This globular configuration is likely a reflection of underlying biological homogeneity, suggesting that, at a global level, the chromatin accessibility landscapes of the different conditions are broadly similar.

Then, we use the same colour code for each condition as the manuscript, observing that P and G were well-mixed and formed a separate region distinct from GS/GSA. While GA showed an intermediate pattern. GS and GSA conditions grouped closely in a shared region of high cell density (Figure IV, corresponding to the Supplemental Fig. 4A of the revised manuscript).

Figure IV: UMAPs colored by experimental conditions revealed distinct regional distributions.

This suggests condition-specific chromatin landscapes, with GS/GSA exhibiting a unique epigenomic signature, while P and G cluster together.

To further validate the chromatin accessibility patterns observed in the scATAC-seq analysis, we merged the bulk and scATAC-seq data and processed all samples uniformly using the bulk ATAC-seq analysis pipeline, treating each scATAC-seq sample as a pseudo-bulk dataset. This uniform processing strategy ensured direct comparability between the pseudobulked scATAC-seq and bulk ATAC-seq data, eliminating analytical bias introduced by differing pipelines. Downstream analysis demonstrated strong concordance between pseudobulk and bulk ATAC-seq datasets, validating the reliability of our single-cell approach. Notably, the vast majority of chromatin accessibility peaks were shared between bulk and pseudobulk data, confirming that our scATAC-seq analysis faithfully recapitulated the global chromatin landscape (Figure V and Supplementary Fig. 4B of the manuscript). Crucially, condition-specific chromatin accessibility profiles were consistently preserved in the pseudobulked scATAC-seq data, reinforcing both the biological relevance and sensitivity of our single-cell methodology. This robust agreement between bulk and single-cell data is particularly remarkable given the inherent sparsity of scATAC-seq, underscoring the technical rigor of our experimental and computational approaches.

Together, these results reinforce the conclusion that the chromatin accessibility features identified via scATAC-seq are robust and consistent with bulk-level biology. Treating the single-cell data as bulk not only confirmed the reproducibility of key findings, but also strengthened the case that the observed UMAP structure reflects a homogeneous yet subtly variable epigenomic landscape among the different experimental conditions (Figure V of the report, corresponding to the Supplementary Fig. 4B of the revised manuscript).

C
Figure V. Heatmap representing the integration of bulk ATAC seq (bulk) and single cell ATACseq data (sc).

In order to demonstrate that the quality of the single cell ATACseq data is optimal we run additional quality control metrics supporting the integrity of the dataset. TSS enrichment scores were high, indicating a strong signal-to-noise ratio and enrichment at transcription start sites. However, the FRiP (Fraction of Reads in Peaks) values were notably low in all samples, suggesting a relatively small proportion of fragments fell within confidently called accessible regions—an indication that the assay's sensitivity to subtle regulatory changes may be limited (Figure VI).

Figure VI. Quality control and metrics for SHARE-seq library. (*Left*) Histogram showing the distribution of Fraction of Reads in Peaks (FRiP) values across single cells. FRiP measures the proportion of reads mapping to called accessible chromatin regions, reflecting data quality. (*Right*) Transcription Start Site (TSS) enrichment plot for individual cells. Each dot represents a cell, with TSS enrichment score on the y-axis.

Taken together, these findings suggest that the chromatin landscape across conditions is globally similar—consistent with the expected biology of the samples—and that this similarity is preserved even when single-cell data are aggregated and compared to bulk ATAC-seq, reinforcing the validity of the single-cell profiles. Nonetheless, modest, condition-specific differences in accessibility do exist and are detectable, particularly as regional skews within the circular UMAP structure, rather than as distinct clusters. These subtle variations, while biologically meaningful, likely fall below the resolution threshold of standard scATAC-seq due to inherent sparsity and limited signal depth and may therefore be underrepresented in downstream analyses.

4. To further characterize the bulk chromatin landscape of their control and mutant hiPSC lines the authors should consider comparing their data to published bulk ATAC data of hematopoietic progenitor cell populations (e.g. from doi: 10.1038/ng.3646). This may help to contextualize the observed changes.

We thank the reviewer for highlighting this point. Indeed, the experiment suggested it was performed as part of our study; however, the reference cited in the original version of the manuscript was incorrect. We sincerely apologize for this oversight. The reference has now been corrected in the revised version of the manuscript and the figure up-dated (see Supplementary Fig. 3C of the revised manuscript).

5. Regarding Motif analyses in differential ATAC peaks, the authors need to revise their interpretation of motifs. Figure 3C suggests that this method would allow the identification of particular transcription factors. However, it should be clear that, based on motif correlation, one cannot say that GATA1 or GATA2 binding is changed. Equally, MEIS2 or HOXA9 cannot really be distinguished from HOXC9 or MEIS1, because they essentially all bind the same motifs (in case of MEIS even only indirectly through a homeobox family member). Motifs generally refer to a family of TF with similar DNA-binding domains, rather than particular family members. Pinning down particular factors would require additional, more specific data such as ChIP or Cut&Tag/Run, etc.

We thank the Reviewer for pointing out this lack of precision. We have now corrected the text accordingly in the new version of the manuscript. As the reviewer can appreciate in the revised version of the manuscript new analysis has been performed including GA condition. This new analysis allowed for a deeper DNA motif enrichment analysis. The results validate our previous data and confirm GATA motifs as the most prevalent motifs in closed chromatin regions across GA, GS, and GSA conditions.

Other comments:

1. There is a discrepancy between Figure 3C and Figure S2D regarding significance values of PU.1. In the supplementary Figure S2D, some P values are reported as 1.00E-02 (including PU.1 in the top panel). However, this P value is not considered significant, since random motif searches can produce much lower P values. If these are typos, they should be corrected. If not, motifs with low enrichment P values should be removed.

We thank the Reviewer for spotting this oversight. A deeper DNA motif enrichment analysis has been used, the findMotifsGenome.pl function implemented in HOMER was used with default parameters to identify statistically enriched transcription factor binding motifs within the DAPs chromatin regions, using the hg38 assembly as reference genome. This analysis has been included in the new version of the manuscript (Figure 3C).

2. Some of the experimental procedures are missing, such as motif analyses.

We thank the Reviewer for pointing out this lack of this information. We have now better include in the section of Math&Methods a more detailed description of our experimental procedures including that of motif analysis.

RE: NCOMMS-24-494668A

Point-by-point response to the referees' comments

Reviewer #1 (Remarks to the Author):

The authors have conducted a significant amount of additional experiments to address all of the reviewers comments. The results of these additional experiments have strengthened the manuscript. I have no additional concerns. This is a very strong study.

We thank the reviewer for the positive feedback and for acknowledging our work. We are pleased to hear that the additional experiments strengthened the manuscript.

Reviewer #2 (Remarks to the Author):

My criticisms were successfully dealt with in the revised version of the manuscript.

Thank you for your time and effort in reviewing our manuscript. We appreciate your insightful comments throughout the revision process and are pleased to hear that you found our responses and the revised version to have successfully addressed your criticisms.

Reviewer #3 (Remarks to the Author):

The revised manuscript addresses major issues present in the previous version, includes data for the previously missing control cell lines, and incorporates single-cell ATAC-seq (scATAC) in addition to bulk ATAC-seq, as recommended by this reviewer. The study design has improved substantially; however, there remains room for further enhancement, particularly in the analysis. Final comments:

We thank the reviewer for him/her time and continued engagement with our manuscript. We have thoroughly considered and addressed their latest points.

1. To ensure reproducibility of computational methods, please provide the exact versions of all software packages and genomes used.

We apologize for the previous lack of detail in our description of the computational methods. This information has now been added in the revised version of the manuscript and is highlighted in yellow (See lines 566-570 and 752).

2. The authors selectively provide numbers for differentially expressed genes and differentially accessible regions. Including all results in supplementary tables would greatly benefit readers.

We thank the reviewer for this insightful and constructive suggestion. We have now added in the new version of the manuscript three additional supplementary tables: Supplementary table 1 containing differential accessible peaks (DAPs) from bulk ATAC-seq, Supplementary table 2 containing DAPs from scATAC-seq, Supplementary table 3 with DAPs from pseudobulk and bulk ATAC-seq integration, and Supplementary table 6 with differential expressed genes (DEGs) from RNA-seq.

3. In the analysis of the ATAC data, the authors are selective about which types of comparisons and datasets they present. Why are the S and A conditions excluded from the genome tracks and other analyses?

We thank the reviewer for this observation. First, we would like to clarify that we did not exclude the S and A conditions from our analysis. As cited in the manuscript (See lines 220-223), we included all DAPs of S vs P and A vs P that were subsequently maintained in the context of GATA2 deficiency (GA, GS, and GSA conditions). This is consistent with the main aim of our study, which is to investigate the effect of these somatic mutations in a GATA2-deficient background. In response to the reviewer's suggestion, we have now provided a clearer presentation of the S vs P and A vs P datasets, including genome tracks (Figure 3A, Figure 3B, Supplemental Figure 4A, Supplemental Figure 4B), motif analyses (Supplemental Figure 6A, Supplemental Figure 6B), and volcano/MvA plots (Supplemental Figure 3A, Supplemental Figure 3B) in the revised version of the manuscript.

4. For their differential calling of accessible regions, the authors use a relatively small logFC cutoff of 0.5, which corresponds to a fold change of approximately 1.4. This threshold is low for ATAC-seq data and may lead to the inclusion of normalization artifacts. It would be helpful if MvA plots for all comparisons were provided in the supplement, highlighting the differentially accessible peaks (DAPs).

We thank the Reviewer for this constructive suggestion. Respectfully, we would like to clarify our rationale for choosing a log₂ fold change (log₂FC) cutoff of |0.5|. This threshold was selected to ensure high sensitivity, allowing us to capture biologically relevant, albeit potentially modest, effects that might otherwise be overlooked. Importantly, as shown in the new Supplementary Table 1, although we applied a log₂FC cutoff of |0.5|, most of statistically significant DAPs (adjusted p < 0.05) show robust effect sizes: [~17%] display a |log₂FC| between 0.7–1, and, notably, [~71%] exceed |log₂FC| > 1. Only [~11%] of significant DAPs fall within the range of 0.5–0.7. A summary of this distribution is provided in Table I. These results demonstrate that our findings are not driven by normalization artifacts but are instead supported by strong and biologically meaningful changes.

LogFC	0.5-0.7	0.7-1	>1
GSA	11%	18%	71%
GS	11%	20%	69%
G	17%	17%	66%
GA	10%	13%	77%
S	14%	21%	64%
A	7%	12%	80%

Table I. Percentage of genes classified per log₂FC intervals, |log₂FC| 0.5–0.7, |log₂FC| > 0.7–1 and |log₂FC| > 1

Finally, we are grateful to the reviewer for this excellent suggestion. As recommended, we have now included an MvA plot for better visualization of the data, which can be found in revised manuscript Supplemental Figure 3B.

5. The tornado plot in Figure 2B suggests very similar profiles for P, G, GS, and GSA, which is unexpected given the large number of comparisons. Three K-means clusters are likely insufficient to distinguish sample-specific or group-specific differences. How

many regions are shown in total? Currently, it is unclear what this plot is intended to demonstrate.

Thank you for this insightful comment and for highlighting the lack of clarity in the original figure. You are correct that the initial analysis with three K-means clusters was insufficient to resolve the subtler group-specific differences, resulting in the overlapping profiles for P, G, GS, and GSA. We have now increased the number of K-means clusters to 5. This refined analysis distinguishes the sample groups more effectively, and the group-specific differences are now more clearly appreciable in the revised Figure 2B.

To answer your specific question, the tornado plot represents the analysis of **7089 genomic regions** in total. The primary intention of this figure is to provide a high-level overview of the variance in epigenetic profiles across our experimental conditions, demonstrating both the major shared patterns (captured by the clusters) and the distinct, group-specific signatures that emerge upon deeper analysis.

Thank you again for prompting this important improvement.

6. The nature of DAPs is not always clear. For example, in the legend of Figure 3C, the authors state, “The top 25 enriched transcription factor (TF) binding motifs are shown for DAPs across GA (left), GS (middle), and GSA (right) conditions,” but it is unclear relative to which condition this enrichment is measured. Additionally, why aren’t DAPs shown for other conditions (e.g. in the supplement)?

We thank the Reviewer for identifying this omission. We have now clarified in both the main figure and its corresponding legend that all comparisons were performed using the Parental line as the reference. Furthermore, we have included the comparisons for S vs P and A vs P in Supplementary Figure 6.

7. Currently, the authors make very limited use of the scATAC data. In their rebuttal letter, they argue that “these results reinforce the conclusion that the chromatin accessibility features identified via scATAC-seq are robust and consistent with bulk-level biology. Treating the single-cell data as bulk not only confirmed the reproducibility of key findings but also strengthened the case that the observed UMAP structure reflects a homogeneous yet subtly variable epigenomic landscape among the different experimental conditions (Figure V of the report, corresponding to the Supplementary Fig. 4B of the revised manuscript). “However, simply comparing pseudo-bulk with bulk data from the same samples does not demonstrate homogenous populations. They should overlap because both represent population averages. While dimensionality reduction of the single-cell data does not separate cells into distinct clusters, it clearly structures the data, with some clusters containing variable contributions from different cell lines, suggesting a degree of heterogeneity within the lines. What is clearly missing is an analysis of chromatin profiles across the observed clusters, including motif searches (e.g., using ChromVAR or similar tools) in differentially accessible regions. Including example tracks of the pseudo-bulk data may also be helpful.

We sincerely thank the reviewer for this insightful comment. We respectfully would like to point out that we do not dispute the potential existence of underlying biological differences. However, our previous and following in depth analyses (described below) indicate that the scATAC-seq data, in this instance, lacked the statistical power and resolution to confidently discriminate between the experimental conditions. This limitation is attributable to the well-documented technical constraints of the method,

primarily the characteristic sparsity of the data and the associated challenges of generating deeply sequenced libraries from limited starting material, which collectively constrain the detection of more subtle differential accessibility. Nevertheless, to directly address the Reviewer's concern, we performed complementary analyses.

Specifically, to evaluate whether the apparent shift in the GS and GSA condition (observed in the UMAP plots included in the previous version of the manuscript as Supplementary Figure 4A) reflect a degree of heterogeneity within the lines or true biological differences, we re-analyzed the scATAC-seq data by processing each experimental condition separately (Figure A). This analysis does not provide a strong signal to separate the cells into distinct groups, indicating that the chromatin landscape is **largely uniform**. A likely biological explanation is that the cell population analyzed is sufficiently homogeneous, which does not allow us to detect epigenetic changes within the different conditions.

Figure A. Dimensionality reduction representation of the chromatin accessibility per condition; Parental, P; GATA2, G; GATA2/ASXL1 (GA); GATA2/SETBP1 (GS) and GATA2/SETBP1/ASXL1 (GSA)

Furthermore, guided by the advice of experts in single-cell genomics (Dr. MA. Mulaw - Ulm University and Dr. J. Pozniak- KU Leuven), we performed two additional quantitative analyses to assess cluster quality and cell state heterogeneity (now included as Supplemental Figure 5B-C). First, we measured the Shannon entropy of each cell, which shows that all conditions have a comparable profile of chromatin accessibility (Figure B left panel). Second, we calculated the Silhouette width, a metric that evaluates how well an individual cell fits to its assigned cluster. The resulting values were consistently close

to 0 across all conditions (Figure B right panel), indicating the chromatin landscape is **largely uniform**, consistent with the analysis described in Figure A.

Figure B. Entropy and Silhouette width plot of the conditions Parental, P; GATA2, G; GATA2/ASXL1 (GA); GATA2/SETBP1 (GS) and GATA2/SETBP1/ASXL1 (GSA).

We agreed that the UMAP projection, as previously shown in Supplementary Figure 4A, apparently organizes the cells into discernible structures across the lines. However, upon a detailed re-examination at lower resolution, these patterns are likely due to an over-clustering effect. In fact, as it can be appreciated in the Figure C, when we generate the UMAP at 0.3 resolution, the over-clustering persists with numerous sub-clusters composed of only a few cells each (singletons).

Figure C. UMAP plot at 0.5 and 0.4 and 0.3 resolution

Overall, these first set of new analysis point towards the absence of major substructures within our populations across the different cell lines.

Although our scATAC-seq appear to lack the resolution to detect statistically significant chromatin accessibility differences between the specific conditions, we nevertheless performed a differential accessibility analysis between Parental (P) condition versus GATA2 (G), GATA2/ASXL1 (GA), GATA2/SETBP1 (GS), and GATA2/SETBP1/ASXL1 (GSA) to identify genomic regions that are significantly more open or more closed. However, we only found 986 DAPs in total (Table II). After annotating regions to their

nearest genes, we found that most of the associated genes (n=474, Table III) were not biologically relevant for our study.

DAPs	G vs P	GA vs P	GS vs P	GSA vs P	
OPEN	75	95	218	103	
CLOSED	132	128	156	79	Total DAPs
Total per condition	207	223	374	182	986

Table II. Differential accessible peaks (DAPs) between the experimental conditions.

Genes	G vs P	GA vs P	GS vs P	GSA vs P	
OPEN	31	41	155	58	
CLOSED	54	48	72	25	Total genes
Total per condition	85	89	227	73	474

Table III. Genes associated to the DAPs summarized in the Table III.

Finally, we performed a ChromVAR analysis, which identified condition-specific enrichment of distinct TF families, among them GATA, HOX, JUN:FOS in line with our previous analysis of bulk ATAC-seq (Table IV and Table V). However, due to the inherent limitations of the methodology mentioned before, we intentionally did not base any major conclusions on this data to ensure the robustness of our interpretation.

More accessible Chromatin Top25 TFs
 p-val adj:<0,01
 avg log2|FC|:>0,5

GATA2 (G)		GATA2/ASXL1 (GA)		GATA2/SETBP1 (GS)		GATA2/SETBP1/ASXL1 (GSA)	
SPIB	IRF1	FOS::JUN	BATF3	MEIS2(var.2)	HOXC10	MEIS2(var.2)	HOXC9
STAT1::STAT2	JUN::JUNB	FOSL2::JUND	BATF::JUN	PBX2	RFX2	PBX2	ETV4
SPI1	BACH2	FOSL2::JUNB	BATF	MEIS1(var.2)	RFX5	MEIS1(var.2)	HOXD12
PLAGL2	IRF2	FOSB::JUNB	JUN(var.2)	HOXD9	HOXA10	HOXA10	HOXD4
		FOS	FOSL1::JUNB	TWIST1	CDX2	HOXD9	HOXB9
		FOS::JUNB	FOSL1	CDX1	NEUROG2(var.2)	CDX1	GABPA
		BACH2	FOS::JUND	CDX4	HOXA13	ZBTB18	TAL1::TCF3
		FOSL1::JUN	NFE2	RFX4	HOXC9	CDX4	HOXC11
		JUNB	JUN::JUNB	ZBTB18	HOXC11	CDX2	ELF1
		JUND	FOSL2	RFX3	HOXB9	HOXA13	MNX1
		FOSL2::JUN	BACH1	RUNX2	HOXD10	HOXC10	ETV1
		JDP2	MAFK	CTCF	NEUROD1	TWIST1	HOXC13
		FOSL1::JUND		RFX1		NEUROG2(var.2)	

Table IV. TOP25 Transcription factors identify using ChromVAR FC>0,5; pval adj<0,01

Less accessible Chromatin Top25 TFs
p-val adj:<0,01
avg log2|FC|:>0,5

GATA2 (G)		GATA2/ASXL1 (GA)		GATA2/SETBP1 (GS)		GATA2/SETBP1/ASXL1 (GSA)	
GATA3	SP1	KLF4	E2F6	KLF6	IRF1	CEBPA	STAT1::STAT2
GATA6	KLF2	KLF5	KLF14	SP4	SP9	CEBPD	IRF6
GATA4	KLF14	SP9	KLF2	KLF4	STAT1::STAT2	NFIL3	HLF
GATA2	KLF3	KLF16	GATA5	ZNF460	SP3	CEBPB	MEF2C
GATA1::TAL1	KLF5	SP3	GATA3	KLF2	CEBPD	CEBPG	ATF4
GATA5	KLF16	ZNF148	GATA4	SP1	FOS	CEBPE	MEF2A
PBX2	GATA1	SP8	GATA6	CEBPG(var.2)	HLF	CEBPG(var.2)	MEF2D
MEIS2(var.2)	SP2	KLF15	CTCF	CEBPA	CEBPB	IRF1	NFE2L1
SP4	MEIS1(var.2)	SP2	KLF9	KLF16	NFIL3		
SP9	KLF11	ZNF263	GATA2	MAF::NFE2	CEBPG		
SP3	KLF4	SP1	MAZ	KLF15	FOS::JUND		
KLF15	SP8	SP4	KLF3	FOSL2	SPI1		
SP1		KLF11		MAFK			

Table V. TOP25 Transcription factors identify using ChromVAR FC<-0,5; pval adj<0,01

We would like to emphasize that the scATAC-seq analysis represented the most challenging aspect of the revision. We have invested significant effort in optimizing every stage of our single-cell analysis, from the experimental protocol to the bioinformatics pipeline, to guarantee robust and high-quality data. However, due to technical constraints inherent to the method, our scATAC-seq dataset appears to lack the resolution required to robustly detect statistically significant differences in chromatin accessibility between specific conditions. For this reason, we present the single-cell data as a pseudo-bulk profile, allowing the integration with our robust bulk ATACseq data, an approach that, in our view, strengthens the manuscript. This analysis draws comparative conclusions while transparently acknowledging the technical limitations of the single-cell data in our system. We hope that the reviewer will appreciate our commitment to methodological rigor.

8. *Example tracks comparing bulk and pseudo-bulk in Suppl. Fig. 4B would also be helpful to judge the quality and comparability of the data.*

We are grateful to the reviewer for raising this important point. To allow for a direct visual assessment of the quality and comparability between our bulk and pseudo-bulk datasets, we have followed the suggestion and added example genome browser tracks. These can now be found in Supplemental Figure 5E-F.

9. *Overlays with published ChIP-seq for GATA2 are used to define GATA2 targets. While this is a possible way to approximate GATA binding sites, the availability of ATAC data also allows to directly analyse and quantify GATA footprints within peak regions (e.g. using Tobias for bulk, ChromVAR for scATAC data or similar tools). This may be more accurate in defining differences between conditions.*

We thank the Reviewer for this valuable suggestion. To clarify, we had already compared the DAPs with genomic regions from the ChIP-seq data. Genomic intervals present in both datasets were defined as GATA2 target loci. This integrative methodology served to validate the findings by not only confirming the presence of a canonical GATA binding motif *in silico* but, more importantly, by providing direct experimental evidence that GATA2 is bound at that specific genomic location. In addition, as the reviewer suggested, we evaluated the utility of TOBIAS for TF footprinting analysis on our ATAC-seq data. We implemented the complete pipeline, which includes ATAC-seq bias correction,

footprint score calculation, and subsequent BINDetect analysis for pairwise comparisons against progenitor conditions.

However, this analysis predicted only a limited number of GATA2 binding events (approximately 4,000 sites per comparison). The output of the analysis is substantially lower than the >21,000 high-confidence sites identified in the GATA2 ChIP-seq published by Fujiwara et al. (2009). Consequently, the overlap between these TOBIAS-predicted sites and our differentially accessible peaks (DAPs) was exceedingly sparse, rendering the comparative analysis biologically uninformative.

We therefore conclude that, while a valuable tool in principle, the TOBIAS algorithm, when applied to our specific dataset, appears to significantly underestimate true GATA2 occupancy compared to established experimental benchmarks.

To provide a more comprehensive view, we employed the FIMO (Find Individual Motif Occurrences) tool to perform an independent scan for GATA family binding motifs within our set of DAPs. This *de novo* motif-based analysis yielded results that were highly consistent with the intersections derived from the ChIP-seq dataset. The strong concordance between these two distinct methodological approaches significantly reinforces the robustness and validity of our initial integrative strategy.

FIMO identified a canonical GATA motif in 15% (106/687) of peaks that gained accessibility (open DAPs). In contrast, the same motif was significantly more prevalent enriched in 35% (251/714) of peaks that lost accessibility (closed DAPs) (Supplementary Table 5).

In summary, while *in silico* footprinting tools like TOBIAS are valuable for predicting transcription factor binding, our results with experimental ChIP-seq data was a more reliable and comprehensive method to identify high-confidence GATA2 binding loci.

10. I did not find any information on how the categorization of regulatory elements e.g. in Fig 3A&B was done. E.g. what's the evidence for enhancers to be "strong"?

Thank you for this critical question, and we apologize for the lack of specificity in our original manuscript regarding the categorization of regulatory elements.

The categorization of enhancers (e.g., "strong") was performed using the standard **GeneHancer track** (version 2) on the **UCSC Genome Browser** (db=hg38). We have now explicitly stated this in the Figure 3 legend and the Methods section.

To answer your specific question: within the GeneHancer database, an enhancer is not defined by a single metric but by a combination of computational and evidence-based factors integrated into a single **"GH Score."**

A "strong" enhancer is therefore operationally defined as a region with a **high GH score** (we applied a threshold of > 50), indicating robust support from numerous independent datasets.

We have now added a summary of this definition in methods section, and it is mentioned in the figure 3 legend to ensure complete transparency. Thank you again for prompting this essential clarification.

11. In the discussion readers should be reminded that the ASXL1 mutation corresponds to a loss-of-function, while the SETBP1 mutation represents a gain-of-function.

We thank the reviewer for the constructive comment. We have now added in the new version of the manuscript this information (See line 377).

12. Some of the current discussion comparing the two mutations was difficult to follow. E.g. "line 381: Importantly, the acquisition of SETBP1 mutation was associated with the most stable and profound chromatin accessibility changes, while ASXL1-induced alterations were largely transient and did not persist into later disease stages." How did

the authors define changes as being transient or non-persistent, if they only have a single time-point?

We thank the Reviewer for this important point. By describing ASXL1-induced changes as 'transient' or 'non-persistent,' do not mean that they were monitored across multiple time-points. Rather we refer to the fact that the chromatin accessibility alterations observed upon acquisition of the ASXL1 mutation are lost once the cells subsequently acquire the SETBP1 mutation. In contrast, the accessibility changes associated with SETBP1 are maintained in the GSA condition, suggesting a more dominant effect. We have now changed the sentence accordingly in the new version of the manuscript with the following sentence: "Importantly, in the context of GATA2 deficiency, the acquisition of the SETBP1 mutation appeared to exert a more dominant impact than ASXL1, as the chromatin accessibility changes observed in the GS condition were largely preserved in GSA, whereas those induced by ASXL1 in GA were mostly lost in the presence of SETBP1" (See lines 394-397).